# Convergence Of Consistency Model With Multistep Sampling Under General Data Assumptions

## Abstract

Diffusion models accomplish remarkable success in data generation tasks across various domains. However, the iterative sampling process is computationally expensive. Consistency models are proposed to learn consistency functions to map from noise to data directly, which allows one-step fast data generation and multi-step sampling to improve sample quality. In this paper, we study the convergence of consistency models when the self-consistency property holds approximately under the training distribution. Our analysis requires only mild data assumption and applies to a family of forward processes. When the target data distribution has bounded support or has tails that decay sufficiently fast, we show that the samples generated by the consistency model are close to the target distribution in Wasserstein distance; when the target distribution satisfies some smoothness assumption, we show that with an additional perturbation step for smoothing, the generated samples are close to the target distribution in total variation distance. We provide two case studies with commonly chosen forward processes to demonstrate the benefit of multistep sampling.

## 1 Introduction

Diffusion models have been widely acknowledged for their high performance across various domains, such as material and drug design (Xu et al., 2022; Yang et al., 2023; Xu et al., 2023), control (Janner et al., 2022), and text-to-image generation (Black et al., 2023; Oertell et al., 2024). The key idea of diffusion models is to transform noise into approximate samples from the target data distribution by iterative denoising. This iterative sampling process typically involves numerical solutions of SDE or ODE, which is computationally expensive especially when generating high-resolution images (Song & Dhariwal, 2024; Ho et al., 2020; Song et al., 2021; Zhang & Chen, 2023; Lu et al., 2022).

Consistency model (CM) (Song et al., 2023) is proposed to accelerate sample generation by learning a consistency function that maps from noise to data directly. It allows both one-step fast data generation and multistep sampling to trade computation for sample quality. Consistency model can be trained with *consistency distillation* or *consistency training* (Song et al., 2023), which enforce that any points on the same trajectory specified by the probability-flow ODE are mapped to the same origin, i.e. the *self-consistency* property. Despite the empirical success of consistency models, little is understood from a theoretical perspective. In particular, recent studies (Luo et al., 2023; Song & Dhariwal, 2024; Kim et al., 2024) observe diminishing improvements in sample quality when increasing the number of steps in multistep sampling. In particular, they find that two-step generation enhances the sample quality considerably while additional sampling steps provide minimal improvements. Such phenomenon motivates the theoretical understanding on consistency models, especially on multistep sampling.

The analysis of consistency models can be challenging for the following reasons:

**Mismatch on the initial starting distributions:** Consistency models generate samples from Gaussian noise (Song et al., 2023) while the ground truth reverse processes (i.e., the denoising process) start from the marginal distribution of the forward process, which is unknown in practice. As

a consequence, we need to analyze the error caused by the mismatch in starting distributions. This difficulty shows up even if we have access to the ground truth consistency function: the consistency function is not Lipschitz even for distributions as simple as Bernoulli, which makes it challenging to analyze this error pointwise. Because the consistency function is the solution to the probability flow ODE, it is natural to consider the stability of the initial value problem. However, without a strong assumption on the consistency function, this approach results in an upper bound with exponential dependency in problem parameters.

**Approximate self-consistency:**   While the training process enforces the self-consistency property, it is impractical to obtain a consistency function estimate with the point-wise exact self-consistency due to various error sources during training (e.g., optimization errors, statistical errors from finite training examples). It is thus natural to focus on the case where the consistency estimator only has approximate self-consistency under the training distribution. The key challenge is how to transfer the approximate self-consistency measured under the training distribution to the quality of the generated samples (e.g., Wasserstein distance between the learned distribution and the ground truth distribution).

**Complexity of multistep sampling:**   We still have a limited understanding of the theoretical benefit of performing multistep sampling in the inference procedure of CM. When performing multistep inference, we need to apply the consistency estimator to the distributions that are different from its original training distribution. Since we can only guarantee approximate self-consistency under the training distribution, analyzing the benefit of multistep sampling requires us to carefully bound the divergence between the training distributions and the test distributions where consistency estimator will be applied during inference time.

## 1.1   OUR CONTRIBUTIONS

We summarize our contribution as follows:

- Our sets of main theorems establish guarantees for *multistep sampling* with a *general set of forward processes* and an *approximate self-consistent* consistency function estimator. Our results apply to data distribution with *mild assumptions*;

- We provide sample quality guarantees in *Wasserstein* distance for a *general set of forward processes* when the data distribution has *bounded support* or has *light tail*. This result naturally applies to multimodal distributions like Bernoulli. Our result in Wasserstein distance is dimension-free due to a more careful convergence analysis for the forward process;

- Sample quality guarantee in *total variation* distance is established for a *general set of forward processes* when the data distribution satisfies some *smoothness assumption*. In this setting, we utilize an additional smoothing step to translate from Wasserstein distance to total variation distance;

- We conduct two *case studies* to illustrate the implication of our main results on *multistep sampling*. We demonstrate that when using the Ornstein-Uhlenbeck (OU) process as the forward process, two-step sampling can significantly improve the quality of the generated samples in terms of Wasserstein distance to the data distribution under certain conditions. On the other hand, our results indicate that increasing the number of sampling steps beyond two has a limited gain, which is consistent with the empirical findings of CM.

## 1.2   RELATED WORK

The theory of diffusion models has been widely studied. Chen et al. (2023b), Lee et al. (2023), and Chen et al. (2023a) study the convergence of score-based generative model and provide polynomial guarantees without assuming log-concavity or a functional inequality on the data. Our data assumption is similar to that of Lee et al. (2023), which is quite minimal. Recently, deterministic samplers with probability-flow ODE have been explored from the theoretical perspective (Chen et al., 2024; Li et al., 2024a; 2023).

Consistency model, which learns a direct mapping from noise to data via trajectory of probability-flow ODE, is proposed to accelerate the sampling step (Song et al., 2023). Song et al. (2023) provides asymptotic theoretical results on consistency models. At a high level, they show that if

the consistency distillation objective is minimized, then the consistency function estimate is close to the ground truth. However, they assume the consistency function estimator achieves exact self-consistency in a point-wise manner. Such a point-wise accurate assumption is not realistic and cannot even be achieved in a standard supervised learning setting.

Lyu et al. (2023), Li et al. (2024b), and Dou et al. (2024) provide the first set of theoretical results towards understanding consistency models. Lyu et al. (2023) shows that with small consistency loss, consistency model generates samples that are close to the target data distribution in Wasserstein distance or in total variation distance after modification. Li et al. (2024b) focuses on consistency training. Dou et al. (2024) provides the first set of statistical theory for consistency models. However, we notice that all of these works require a strong assumption on the data distribution. Specifically, they assume that the ground truth consistency function is Lipschitz. While the Lipschitz condition allows a direct approach to control the error of mismatch on the initial starting distribution, it's unclear how large the Lipschitz coefficient is. A direct application of Gronwall's inequality typically results in a Lipschitz constant with exponential dependency on problem parameters. To overcome this, we use the data-processing inequality, which only requires approximate self-consistency and minor assumptions on target data distribution. Moreover, our upper bound is polynomial in problem parameters.

## 2 PRELIMINARIES

Score-based generative models (Song et al., 2021) and consistency models (Song et al., 2023) aim to sample from an unknown *data distribution* $P_{\text{data}}$ in $\mathbb{R}^d$. We review some basic concepts and introduce relevant notations in this section.

**Score-based generative model:** A score-based generative model, or diffusion model (Ho et al., 2020; Song et al., 2021) defines a *forward process* $\{\mathbf{x}_t\}_{t \in [0,T]}$ by injecting Gaussian noise into the data distribution $P_{\text{data}}$ in $d$-dimensional space $\mathbb{R}^d$, where $\mathbf{x}_0 \sim P_{\text{data}}$ and $T > 0$. In this paper, we focus on a general family of forward processes characterized by stochastic differential equations (SDEs) with the following form:

$$d\mathbf{x}_t = h(t)\mathbf{x}_t dt + g(t)d\mathbf{w}_t, \quad \mathbf{x}_0 \sim P_{\text{data}}, \tag{1}$$

where $\mathbf{w}_t$ is the standard Wiener process. It is known that the marginal distribution of $\mathbf{x}_t$ in (1) is Gaussian conditioning on $\mathbf{x}_0$ (Kingma et al., 2021; Lu et al., 2022):

$$\mathbf{x}_t | \mathbf{x}_0 \sim \mathcal{N}(\alpha_t \mathbf{x}_0, \sigma_t^2 I), \quad \forall t \in [0, T],$$

where $\alpha_t, \sigma_t \in \mathbb{R}^+$ is specified by $h(t) = \frac{d \log \alpha_t}{dt}$, $g^2(t) = \frac{d\sigma_t^2}{dt} - 2\frac{d \log \alpha_t}{dt}\sigma_t^2$ with proper initial conditions. $\alpha_t$ and $\sigma_t^2$ specify the *noise schedule* of the forward process. The noise schedule $\{(\alpha_t, \sigma_t^2)\}_{t \in [0,T]}$ and initial data distribution determine the *marginal distribution* of the forward process $\{P_t\} \in [0, T]$, where $\mathbf{x}_t \sim P_t$ and $P_0 = P_{\text{data}}$. We use $\{p_t\}_{t \in [0,T]}$ to denote the *probability density functions* (PDFs) of $\{P_t\}_{t \in [0,T]}$. For simplicity, we use $\mathcal{D}(\cdot; \alpha_t, \sigma_t^2)$ to denote the operator on distributions defined by a noise schedule $(\alpha_t, \sigma_t^2)$. Specifically, given any distribution $P$, $\mathcal{D}(P; \alpha_t, \sigma_t^2)$ is the marginal distribution of $\mathbf{x}'$, where $\mathbf{x}'|\mathbf{x} \sim \mathcal{N}(\alpha_t \mathbf{x}, \sigma_t^2)$ and $\mathbf{x} \sim P$. When it is clear from the context, we use $\mathcal{D}(\cdot, t)$ as a shorthand. With this notation, marginal distribution is expressed as $P_t = \mathcal{D}(P_{\text{data}}, t)$.

The forward process specified by (1) converges to Gaussian distribution $\mathcal{N}(0, \sigma_t^2 I)$ for some properly chosen $h(\cdot)$ and $g(\cdot)$ (Bakry et al., 2014; Song et al., 2021) (interested readers may refer to Lemma 3 for an explicit dependency on the noise schedule). The convergence of the forward process facilitates a procedure to generate samples from $P_{\text{data}}$, approximately: generate a sample from $\mathcal{N}(0, \sigma_T^2 I)$ and feed it to an approximate reversal of (1). However, the reverse-time SDE of (Equation 1) is usually computationally expensive.

It is known that the following *probability flow ordinary differential equation* (PF-ODE) generates the same distributions as the marginals distribution of (1) Song et al. (2021):

$$\frac{d\mathbf{x}_t}{dt} = h(t)\mathbf{x}_t - \frac{1}{2}g^2(t)\nabla \log p_t(\mathbf{x}_t), \quad \mathbf{x}_0 \sim P_{\text{data}}. \tag{2}$$

The time-reversal of (2) defines a deterministic mapping from noise to data, which facilitates *consistency model* (Song et al., 2023) as a computationally efficient one-step sample generation.

**Consistency models:** A consistency model (Song et al., 2023) is an alternative approach to generate samples from $P_{\text{data}}$: instead of solving the reversal of the SDE in (1), one could directly learn a *consistency function* that maps a point on a trajectory of (2) to its origin. For any $\mathbf{x}$ and $t_0 \geq 0$, let $\{\varphi(t; \mathbf{x}, t_0)\}_{t \in [0, T]}$ be the trajectory specified by (2) and *initial condition* $\mathbf{x}_{t_0} = \mathbf{x}$.[1] The (ground truth) consistency function of (2) is defined to be:[2]

$$f^\star(\mathbf{x}, t) := \varphi(0; \mathbf{x}, t), \quad \forall \mathbf{x} \in \mathbb{R}^d, t \geq 0. \tag{3}$$

A consistency function enjoys the *self-consistency* property: if $(\mathbf{x}, t)$ and $(\mathbf{x}', t')$ are on the same trajectory of (2), they are mapped to the same origin, i.e. $f^\star(\mathbf{x}, t) = f^\star(\mathbf{x}', t')$.[3]

The self-consistency property of the ground truth consistency function $f^\star(\cdot, \cdot)$ enlightens the training for consistency function via enforcing the self-consistency property instead of learning the mapping from noise to data directly. At a high level, in the training stage, we first discretize the interval $[0, T]$ with the following partition:

$$\mathcal{T} : 0 = \tau_0 < \tau_1 < \tau_2 < \cdots < \tau_M = T.$$

For simplicity, we assume the partition is equal, i.e. there exists $\Delta\tau > 0$, s.t. $\tau_i = \Delta\tau \cdot i$, for $i = 1, \ldots, M$. We then enforce the self-consistency property on each partition point by finding some $\hat{f}(\cdot, \cdot)$, s.t.

$$\mathbb{E}_{\mathbf{x}_{\tau_i} \sim P_{\tau_i}} \left[ \left\| \hat{f}(\mathbf{x}_{\tau_i}, \tau_i) - \hat{f}(\varphi(\tau_{i+1}; \mathbf{x}_{\tau_i}, \tau_i), \tau_{i+1}) \right\|_2^2 \right] \tag{4}$$

is small for all $i = 0, 1, \ldots, M - 1$. This strategy is justified by our theoretical results in Section 3: *even if the self-consistency property is violated slightly, the consistency function estimation will produce high-quality samples.* In practice, the trajectories of the PF-ODE (2) are unknown, so the self-consistency objective cannot be optimized directly. With this regard, *consistency distillation*, which utilizes a pre-trained score function estimate, and *consistency training*, which builds an unbiased estimate for the score function, are proposed to approximate the transition on the trajectories of the PF-ODE. Interested readers can find the details in Song et al. (2023).

Given a consistency model estimate $\hat{f}(\cdot, \cdot)$, we could generate approximate samples by feeding Gaussian noise into $\hat{f}(\cdot, \cdot)$ using *single-step* or *multistep sampling*. Given $\hat{\mathbf{x}}_T \sim \mathcal{N}(0, \sigma_T^2 I)$, one can generate sample in a single step by calculating $\hat{f}(\hat{\mathbf{x}}_T, T)$. Furthermore, one can also design a sequence of time steps by selecting $N \geq 1$ steps in the training partition $\mathcal{T}$:

$$T = t_1 > t_2 > \cdots > t_N > 0, \tag{5}$$

We refer to the sequence $\{t_i\}_{i=1:N} \subseteq \mathcal{T} \setminus \{0\}$ as *sampling time schedule*. Given this sampling time schedule, one can alternatingly denoise by calculating $\hat{\mathbf{x}}_0^{(i)} = \hat{f}(\hat{\mathbf{x}}_{t_i}^{(i)}, t_i)$ and inject noise by drawing $\hat{\mathbf{x}}_{t_{i+1}}^{(i+1)} \sim \mathcal{N}(\alpha_{t_{i+1}} \hat{\mathbf{x}}_0^{(i)}, \sigma_{t_{i+1}}^2 I)$, where $\hat{\mathbf{x}}_{t_1}^{(1)} = \hat{\mathbf{x}}_T \sim \mathcal{N}(0, \sigma_T^2 I)$ and $i = 1, \ldots, N$. The $\hat{\mathbf{x}}_0^{(N)}$ in the last step is the output of the sampling process. We highlight this procedure as follows:

$$\hat{\mathbf{x}}_{t_1}^{(1)} \xrightarrow{\hat{f}(\cdot, t_1)} \hat{\mathbf{x}}_0^{(1)} \xrightarrow{\sim \mathcal{N}(\alpha_{t_2} \hat{\mathbf{x}}_0^{(1)}, \sigma_{t_2}^2 I)} \hat{\mathbf{x}}_{t_2}^{(2)} \xrightarrow{\hat{f}(\cdot, t_2)} \hat{\mathbf{x}}_0^{(2)} \rightarrow \cdots \rightarrow \hat{\mathbf{x}}_{t_N}^{(N)} \xrightarrow{\hat{f}(\cdot, t_N)} \hat{\mathbf{x}}_0^{(N)}.$$

When $N = 1$, this degenerates to single-step sampling. For completeness, we summarize this process in Algorithm 1 in Section A. For a concise presentation, we defines $\left\{\hat{P}_{t_i}\right\}_{i=1:N}$ to be the sequence of marginal distributions of $\{\hat{\mathbf{x}}_{t_i}^{(i)}\}_{i=1:N}$ and define $\left\{\hat{P}_0^{(i)}\right\}_{i=1:N}$ to be the sequence of marginal distributions of $\left\{\hat{\mathbf{x}}_0^{(i)}\right\}_{i=1:N}$. By the definition of multistep sampling, these two sequences of distributions evolve according to the following recursion and output $\hat{P}_{t_N}$ at the end:

$$\hat{P}_{t_1} = \mathcal{N}(0, \sigma_{t_1}^2); \ \hat{P}_0^{(i)} = \hat{f}(\hat{P}_{t_i}, t_i), \ \hat{P}_{t_{i+1}} = \mathcal{D}\left(\hat{P}_0^{(i)}, t_{i+1}\right), \ i = 1, \ldots, N, \tag{6}$$

---

[1]Specifically, $\varphi(\cdot; \mathbf{x}, t_0)$ is the solution to the ODE *initial value problem* specified by (2) and $\mathbf{x}_{t_0} = \mathbf{x}$

[2]Song et al. (2023) stops at time $t = \delta$ for some small $\delta > 0$ and accepts $\hat{f}(\mathbf{x}, t) = \hat{\varphi}(\delta; \mathbf{x}, t)$, an estimate for $\varphi(\delta; \mathbf{x}, t)$ as the approximate samples to avoid numerical instability. In this paper, we ignore this numerical issue to obtain a cleaner theoretical analysis.

[3]At a high level, this can be shown by contradiction: suppose $(\mathbf{x}', t')$ lies on the trajectory of $(\mathbf{x}, t)$, meaning $\varphi(\cdot; \mathbf{x}, t)$, the trajectory of $(\mathbf{x}, t)$ and $\varphi(\cdot; \mathbf{x}', t')$, the trajectory of $(\mathbf{x}', t')$ intersect at $(\mathbf{x}', t')$. Then both trajectories satisfy the initial condition that takes value $\mathbf{x}'$ at time $t'$. By Picard's existence and uniqueness theorem, the trajectories of $\varphi(\cdot; \mathbf{x}, t)$ and $\varphi(\cdot; \mathbf{x}', t')$ are identical and have the same origin.

where we reuse $\hat{f}(\cdot, \cdot)$ for operation on distributions. Specifically, for any distribution $P$ and $t \geq 0$, we use $\hat{f}(P, t)$ to denote the distribution of $\hat{f}(\mathbf{x}, t)$ when $\mathbf{x} \sim P$. In Section 3, we study *how multistep sampling influences the sample quality* from the theoretical perspective.

**Performance metric:** In this paper, we study the *sample quality* generated by a consistency function estimate $\hat{f}(\cdot, \cdot)$ and the multistep sampling procedure introduced above. To quantify the sample quality, we establish upper bounds on 2-Wasserstein distance ($W_2$) in Euclidean norm, and upper bounds on Total Variation (TV) distance. The 2-Wasserstein distance between two distributions $P$ and $Q$ is defined to be:

$$W_2(P, Q) := \inf_{\gamma \in \Gamma(P,Q)} \sqrt{\mathbb{E}_{(\mathbf{x},\mathbf{y}) \sim \gamma}\left[\|\mathbf{x} - \mathbf{y}\|_2^2\right]},$$

where $\Gamma(P, Q)$ is the set of all joint distributions such that the marginal distribution over the first random variable is $P$ and the marginal distribution over the second random variable is $Q$.

Total Variation distance between two distributions $P$ and $Q$ is defined to be:

$$\mathrm{TV}(P, Q) := \frac{1}{2} \|p(\mathbf{x}) - q(\mathbf{x})\|_1,$$

where $p(\cdot)$ is the PDF of $P$ and $q(\cdot)$ is the PDF of $Q$.

## 3 MAIN RESULTS

In this section, we present theoretical guarantees on sample quality for consistency models with multistep sampling. We first present two sets of results for the *general* forward process in (1) with *arbitrary* sampling time schedule: in Section 3.1, we demonstrate that the generated samples are close to the target data distribution $P_{\text{data}}$ in $W_2$ when $P_{\text{data}}$ has bounded support or satisfies some tail condition; with an additional smoothing step, we show guarantee in TV distance for $P_{\text{data}}$ with smoothness condition in Section 3.2. To illustrate the general results and gain better understanding on the multistep sampling, we choose two special SDEs as forward processes and design sampling time schedules in Section 3.3.

The natural central assumption in our theoretical results is a good consistency function estimate:

**Assumption 1** (A proper consistency model). *Suppose $\hat{f}(\mathbf{x}, 0) = \mathbf{x}$ for all $\mathbf{x} \in \mathbb{R}^d$ and there exists $\epsilon_{\text{cm}} > 0$, s.t. (4) $\leq \epsilon_{\text{cm}}^2$ for all $i = 0, 1, \ldots, M - 1$.*

Firstly, the condition related to the accuracy of the consistency function estimate is necessary: we cannot generate good samples with an arbitrary function. Instead of assuming the output of $\hat{f}(\cdot, \cdot)$ and $f^{\star}(\cdot, \cdot)$ to be close directly, we only require the self-consistency property to hold *approximately* under its training distribution, which aligns with the objective function when training for $\hat{f}(\cdot, \cdot)$. Note that our assumption does not imply $\hat{f}$ will be self-consistent in a point-wise manner.

The self-consistency objective (4) can be approximated via *consistency distillation* or *consistency training* (Song et al., 2023). Consistency distillation uses a pre-trained score function (an estimation for $\nabla \log p_t(\cdot)$) to approximate $\varphi(\cdot; \cdot, \cdot)$ and train for $\hat{f}(\cdot, \cdot)$ with target network and online network. In Section E, we incorporate consistency distillation with minor modifications into our framework without additional data assumptions. On the other hand, consistency training constructs an unbiased estimator for $\nabla \log p_t(\mathbf{x}_t)$ to approximate (4). Theorem 2 of Song et al. (2023) shows that the self-consistency loss (4) can be approximated by consistency training under proper conditions when $\Delta \tau$ is small.

In (4), we use $\|\cdot\|_2^2$ as an error metric, which agrees with the choice in practice Luo et al. (2023); Song et al. (2023). The metric $\|\cdot\|_2^2$ aligns better with the theoretical analysis: on the one hand, Lemma 2 demonstrates that this metric translates naturally to the 2-Wasserstein metric $W_2$; on the other hand, $\|\cdot\|_2^2$ is more suitable for the multi-step sampling because the squared error contracts nicely in the forward process with Gaussian noise as shown by Lemma 1 and 3.

Finally, we remark that Assumption 1 only requires an $L_2$-accurate self-consistent $\hat{f}(\cdot, \cdot)$, instead of requiring $\left\|\hat{f}(\mathbf{x}_{\tau_i}, \tau_i) - \hat{f}(\varphi(\tau_{i+1}; \mathbf{x}_{\tau_i}, \tau_i), \tau_{i+1})\right\|_2$ to be small uniformly for all $\mathbf{x}_{\tau_i}$. On the

empirical side, conditions in $L_2$ norm are more realistic because it allows the approximation to the expectation in (4) with finite data.

## 3.1 GUARANTEES IN WASSERSTEIN METRIC

We now provide upper bounds on the sampling error in $W_2$ distance. We start by considering $P_{\text{data}}$ with bounded support:

**Theorem 1** ($W_2$ error for distributions with bounded support). *Suppose Assumption 1 holds. Suppose there exists $R > 0$, s.t. $\sup_{\mathbf{x}\in\text{supp}(P_{data})}\|\mathbf{x}\|_2 \leq R$ and $\left\|\hat{f}(\mathbf{x},t)\right\|_2 \leq R$ for all $(\mathbf{x},t) \in \mathbb{R}^d \times [0,T]$, Let $\hat{P}_0^{(N)}$ be the output of (6). Then we have:*

$$W_2(\hat{P}_0^{(N)}, P_{data}) \leq 2R\left(\frac{\alpha_{t_1}^2}{4\sigma_{t_1}^2}R^2 + \sum_{j=2}^N \frac{\alpha_{t_j}^2}{4\sigma_{t_j}^2}t_{j-1}^2\frac{\epsilon_{\text{cm}}^2}{\Delta\tau^2}\right)^{1/4} + t_N\frac{\epsilon_{\text{cm}}}{\Delta\tau} \qquad (7)$$

Compared to $P_{\text{data}} = f^\star(P_{t_N}, t_N)$, the sampling error of $\hat{P}_0^{(N)} = \hat{f}(\hat{P}_{t_N}, t_N)$ comes from: (i). starting from a different marginal distribution $\hat{P}_{t_N}$ instead of $P_{t_N}$; (ii). using an inaccurate consistency function estimate $\hat{f}(\cdot,\cdot)$ instead of $f^\star(\cdot,\cdot)$. The term $\frac{\alpha_t^2}{\sigma_t^2}$ characterizes the convergence of   NEW
the forward process as demonstrated by Lemma 3. It converges to 0 quickly for reasonable forward SDE (1). Asymptotically, the right hand side of (7) goes to 0 as $t_1 \to \infty$ and $\epsilon_{\text{cm}} \to 0$.

(7) implies a trade-off when sampling with multiple steps. When using more sampling steps: on one hand, $\frac{\alpha_{t_1}^2}{4\sigma_{t_1}^2}R^2 + \sum_{j=2}^N \frac{\alpha_{t_i}^2}{4\sigma_{t_i}^2}t_{i-1}^2\frac{\epsilon_{\text{cm}}^2}{\Delta\tau^2}$, an upper bound on $\text{KL}(P_{t_N} \parallel \hat{P}_{t_N})$,[4] accumulates; on the other hand, $t_N\frac{\epsilon_{\text{cm}}}{\Delta\tau}$, the error from an inaccurate consistency function decreases due to a shorter $t_N$. The design of sampling time schedule $\{t_i\}_{i=1:N}$, which depends on the noise schedule $\{(\alpha_t, \sigma_t^2)\}_t$, is crucial in achieving good sample quality. We defer design choices for some specific forward processes and simplified upper bounds to Section 3.3.

When $\Delta\tau$ decreases, on the one hand, there would be more intermediate steps in the error decomposition of the consistency function estimate given a fix $t$ (see Lemma 2); on the other hand, using a smaller $\Delta\tau$ allows a smaller $t_N$ and may potentially decrease $\epsilon_{\text{cm}}$ as well. It is challenging to analyze the effect of $\Delta\tau$ quantitatively without further assumption.

The technique in Theorem 1 can be extended to distributions without finite support. When $P_{\text{data}}$ satisfies some tail condition, it is sufficient to sample only from a bounded region:

**Theorem 2** ($W_2$ error for distributions with tail condition). *Suppose there exists $c, C > 0$ and $R \geq C$, s.t. $\text{Pr}_{\mathbf{x}\sim P_{data}}(\|\mathbf{x}\|_2 \geq t) \leq ce^{-t/C}$ for all $t \geq R$. Let $P_{data\cap\mathcal{B}(0,R)}$ be the distribution truncated from $P_{data}$, i.e. the conditional distribution of $\mathbf{x}$ given $\|\mathbf{x}\|_2 \leq R$ where $\mathbf{x} \sim P_{data}$. Let $\varphi_R(\cdot;\cdot,\cdot)$ be the solution to the corresponding PF-ODE and $f_R^\star(\cdot,\cdot)$ be the corresponding consistency function. Let $\{P_t^R\}_{t\in[0,T]}$ be the marginal distribution of the forward process starting from $P_{data\cap\mathcal{B}(0,R)}$. If $\hat{f}(\cdot,\cdot)$ satisfies: (a) $\left\|\hat{f}(\mathbf{x},t)\right\|_2 \leq R$, for all $(\mathbf{x},t) \in \mathbb{R}^d \times [0,T]$; (b) $\hat{f}(\mathbf{x},0) = \mathbf{x}$, for all $\mathbf{x}$;*

*(c) $\mathbb{E}_{\mathbf{x}_t\sim P_{\tau_i}^R}\left[\left\|\hat{f}(\mathbf{x}_t,\tau_i) - \hat{f}(\varphi_R(\tau_{i+1};\mathbf{x}_t,\tau_i),\tau_{i+1})\right\|_2^2\right] \leq \epsilon_{\text{cm}}^2$, for all $i = 0,\ldots,M-1$ for some*

*$\epsilon_{\text{cm}} > 0$. Then $W_2(\hat{P}_0^{(N)}, P_{data}) \leq 2R\left(\frac{\alpha_{t_1}^2}{4\sigma_{t_1}^2}R^2 + \sum_{j=2}^N\frac{\alpha_{t_j}^2}{4\sigma_{t_j}^2}t_{j-1}^2\frac{\epsilon_{\text{cm}}^2}{\Delta\tau^2}\right)^{1/4} + t_N\frac{\epsilon_{\text{cm}}}{\Delta\tau} + O(Re^{-\frac{R}{2C}}).$*

By restricting the output of $\hat{f}(\cdot,\cdot)$ to be $\mathcal{B}(0,R)$, the Euclidean ball with radius $R$, we focus on learning the portion of $P_{\text{data}}$ inside the Euclidean ball. This truncation step reduces the problem of *sampling from unbounded distribution* to *sampling from a distribution with finite support*, at the cost of introducing the additional term $O(Re^{-\frac{R}{2C}})$.

---

[4]We use $\text{KL}(P \parallel Q)$ to denote the Kullback–Leibler (KL) divergence of distribution $P$ from distribution $Q$, which is defined by: $\text{KL}(P \parallel Q) := \int_{\mathbf{x}\in\mathbb{R}^d} p(\mathbf{x}) \log\frac{p(\mathbf{x})}{q(\mathbf{x})}d\mathbf{x}$.

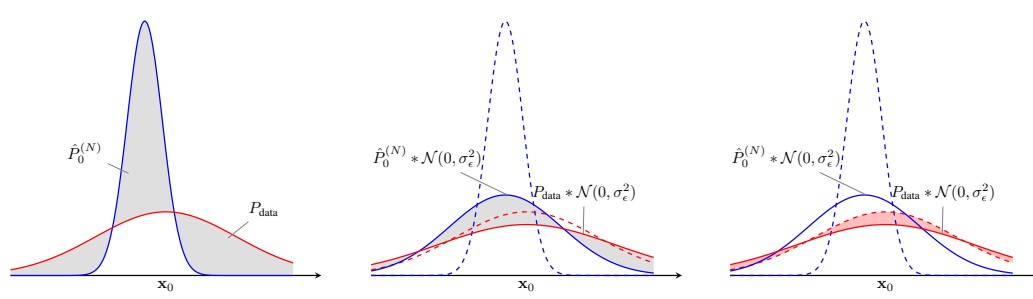

(a) TV distance between $\hat{P}_0^{(N)}$ and $P_{\text{data}}$.

(b) TV distance between $\hat{P}_0^{(N)} * \mathcal{N}(0, \sigma_\epsilon^2 I)$ and $P_{\text{data}} * \mathcal{N}(0, \sigma_\epsilon^2 I)$).

(c) TV distance between $P_{\text{data}} * \mathcal{N}(0, \sigma_\epsilon^2 I)$ and $P_{\text{data}}$.

Figure 1: Smoothing by additional perturbation

### 3.2 GUARANTEE IN TOTAL VARIATION DISTANCE

In the sampling process of consistency models, it is non-trivial to control the error in TV distance. This difficulty arises even when we sample with a single step and can draw samples for the marginal distribution $P_T$ directly. Assumption 1 ensures that $\hat{f}(P_T, T)$ is close to $f^\star(P_T, T)$ in $W_2$. However, $W_2$ and TV have very different structures: $W_2$ controls the pointwise distance between distributions while TV only focuses on the density of the distribution. Even if $W_2(\hat{P}_0^{(N)}, P_{\text{data}})$ is small, the densities of $\hat{f}(P_T, T)$ and $f^\star(P_T, T)$ may not overlap well (see Figure 1a) and $\text{TV}(\hat{f}(P_T, T), f^\star(P_T, T))$ can be as large as 1 if $\hat{f}(P_T, T)$ is nearly deterministic while $f^\star(P_T, T)$ has large variance. As a result, it's not possible to control TV distance only with conditions on $W_2$ distance in general.

One solution is to perturb $\hat{P}_0^{(N)}$ slightly with Gaussian noise $\mathcal{N}(0, \sigma_\epsilon^2)$. With this perturbation, $\hat{P}_0^{(N)} * \mathcal{N}(0, \sigma_\epsilon^2)$ and $P_{\text{data}} * \mathcal{N}(0, \sigma_\epsilon^2)$ could have better overlap and be closer in TV (see Figure 1b), where we use $P * Q$ to denote the *convolution* of distribution $P$ and $Q$. When $P_{\text{data}}$ satisfies smoothness assumption, the perturbation will not change $P_{\text{data}}$ too much so $\text{TV}(P_{\text{data}} * \mathcal{N}(0, \sigma_\epsilon^2), P_{\text{data}})$ is small (See Figure 1c). One could choose a small $\sigma_\epsilon$ and use $\hat{P}_0^{(N)} * \mathcal{N}(0, \sigma_\epsilon^2 I)$ as the output.

**Theorem 3** (TV error for distributions under smoothness assumption). *Suppose Assumption 1 holds. Let $p_{data}(\cdot)$ be the PDF of $P_{data}$. If $\log p_{data}(\cdot)$ is $L$-smooth, then for all $\sigma_\epsilon > 0$, we have:*

$$\text{TV}(\hat{P}_0^{(N)} * \mathcal{N}(0, \sigma_\epsilon^2 I), P_{data}) \leq \sqrt{\frac{\alpha_{t_1}^2}{4\sigma_{t_1}^2} \mathbb{E}_{\mathbf{x} \sim P_{data}} \left[ \|\mathbf{x}\|_2^2 \right] + \sum_{j=2}^{N} \frac{\alpha_{t_j}^2}{4\sigma_{t_j}^2} t_{j-1}^2 \frac{\epsilon_{\text{cm}}^2}{\Delta\tau^2} + \frac{1}{2\sigma_\epsilon} t_N \frac{\epsilon_{\text{cm}}}{\Delta\tau}} + 2dL\sigma_\epsilon.$$

Compared to Theorem 1, the upper bound in Theorem 3 has an additional term $2dL\sigma_\epsilon$. This is the "bias" induced by the additional perturbation $\mathcal{N}(0, \sigma_\epsilon^2 I)$. To get a tighter bound, we may choose $\sigma_\epsilon = \sqrt{\frac{t_N \epsilon_{\text{cm}}}{4dL\Delta\tau}}$, and the upper bound becomes: $\sqrt{\frac{\alpha_{t_1}^2}{4\sigma_{t_1}^2} \mathbb{E}_{\mathbf{x} \sim P_{\text{data}}} \left[ \|\mathbf{x}\|_2^2 \right] + \sum_{j=2}^{N} \frac{\alpha_{t_j}^2}{4\sigma_{t_j}^2} t_{j-1}^2 \frac{\epsilon_{\text{cm}}^2}{\Delta\tau^2}} + 2\sqrt{t_N dL \frac{\epsilon_{\text{cm}}}{\Delta\tau}}$.

### 3.3 CASE STUDIES ON MULTISTEP SAMPLING

To illustrate the theoretical guarantee and understand the benefits of multistep sampling, we conduct case studies with two common forward processes. For simplicity, we assume $P_{\text{data}}$ to have bounded support and ignore the rounding issues when selecting sampling time schedule $\{t_i\}_{i=1:N}$ from the training time partition $\mathcal{T}$.

**Case study 1:** we consider the *Variance Preserving SDE* in Song et al. (2021) with $\beta(t) = 2$ as the forward process:

$$d\mathbf{x}_t = -\mathbf{x}_t dt + \sqrt{2} d\mathbf{w}_t, \quad \mathbf{x}_0 \sim P_{\text{data}}. \tag{8}$$

This is also known as the Ornstein-Uhlenbeck (OU) process and is studied by Chen et al. (2023b) in the context of score-based generative models. The forward process defined by (8) has noise

schedule $\alpha_t = e^{-t}$ and $\sigma_t^2 = 1 - e^{-2t}$ and its marginal distribution is $\mathbf{x}_t \sim \mathcal{N}(e^{-t}\mathbf{x}_0, (1 - e^{-2t})I)$conditioning on $\mathbf{x}_0$. Theorem 1 guarantees:

$$W_2(\hat{P}_{t_N}, P_{\text{data}}) \leq 2R\left(\frac{e^{-2t_1}}{4(1-e^{-2t_1})}R^2 + \sum_{j=2}^{N}\frac{e^{-2t_j}}{4(1-e^{-2t_j})}t_{j-1}^2\frac{\epsilon_{\text{cm}}^2}{\Delta\tau^2}\right)^{1/4} + t_N\frac{\epsilon_{\text{cm}}}{\Delta\tau}. \quad (9)$$

In this case study, we focus on the design of the sampling time schedule based on upper bound (9)/ Surprisingly, we demonstrate that *an ultra-small $t_N$ is not beneficial*. We assume $\tau_1 = \Delta\tau \ll 1$ and $\frac{\epsilon_{\text{cm}}}{\Delta\tau} < R$.[5]

One strategy for designing $\{t_i\}_{i=1:N}$ is to *minimize the upper bound* (9). We first establish a lower bound on (9) as a baseline. Without loss of generality, we assume $t_1 \geq 2$. (9) can be lower bounded as:

$$(9) \geq R\sqrt{\frac{\epsilon_{\text{cm}}}{\Delta\tau}}\left(\sum_{j=2}^{N}\frac{t_j}{e^{2t_j}-1}(t_{j-1}-t_j)\right)^{1/4} + t_N\frac{\epsilon_{\text{cm}}}{\Delta\tau} \geq R\sqrt{\frac{\epsilon_{\text{cm}}}{\Delta\tau}}\left(\int_{t_N}^{2}\frac{x\,\mathrm{d}x}{e^{2x}-1}\right)^{1/4} + t_N\frac{\epsilon_{\text{cm}}}{\Delta\tau},$$

where the first step is because $0 < t_j \leq t_{j-1}$ and the second step is because $\frac{x}{e^{2x}-1}$ monotonically decreases. Let $c_1, c_2 > 0$ be absolute constants, s.t. $\left(\int_{c_1}^{2}\frac{x\,\mathrm{d}x}{e^{2x}-1}\right)^{1/4} = c_2$. Then if $t_N \geq c_1$, $(9) \geq c_1\frac{\epsilon_{\text{cm}}}{\Delta\tau} = \Omega\left(\frac{\epsilon_{\text{cm}}}{\Delta\tau}\right)$; if $t_N < c_1$, $(9) \geq c_2 R\sqrt{\frac{\epsilon_{\text{cm}}}{\Delta\tau}} = \Omega\left(R\sqrt{\frac{\epsilon_{\text{cm}}}{\Delta\tau}}\right)$. In either case, $(9) = \Omega\left(\min\left\{\frac{\epsilon_{\text{cm}}}{\Delta\tau}, R\sqrt{\frac{\epsilon_{\text{cm}}}{\Delta\tau}}\right\}\right)$. The condition $\frac{\epsilon_{\text{cm}}}{\Delta\tau} < R$ further implies $(9) = \Omega\left(\frac{\epsilon_{\text{cm}}}{\Delta\tau}\right)$. Given this lower bound, one heuristic is to set every term in (9) to $\tilde{\Theta}\left(\frac{\epsilon_{\text{cm}}}{\Delta\tau}\right))$ to match this baseline approximately, which requires:

$$t_i \geq \log\frac{R^3\Delta\tau^2}{\epsilon_{\text{cm}}^2}, \text{ if } i = 1; \quad t_i \geq \log\frac{R^2\Delta\tau}{\epsilon_{\text{cm}}}, \text{ o.w..} \quad (10)$$

With this heuristic, a two-step sampling procedure shows an improvement on sample quality:

**Corollary 1** (Two-step sampling with OU process). *Suppose the conditions in Theorem 1 are satisfied. Suppose $\alpha_t = e^{-t}$, $\sigma_t^2 = 1 - e^{-2t}$. Then for $t_1 = \log\frac{R^3\Delta\tau^2}{\epsilon_{\text{cm}}^2}$, $t_2 = \log\frac{R^2\Delta\tau}{\epsilon_{\text{cm}}}$, we have:*

$$\begin{cases} W_2(\hat{P}_0^{(1)}, P_{data}) \leq \frac{\epsilon_{\text{cm}}}{\Delta\tau}\left(\log\frac{R^3\Delta\tau^2}{\epsilon_{\text{cm}}^2} + O(1)\right), \\ W_2(\hat{P}_0^{(2)}, P_{data}) \leq \frac{\epsilon_{\text{cm}}}{\Delta\tau}\left(\log\frac{R^2\Delta\tau}{\epsilon_{\text{cm}}} + O\left(\sqrt{\log\frac{R^2\Delta\tau}{\epsilon_{\text{cm}}}}\right)\right). \end{cases} \quad (11)$$

Because $\frac{\epsilon_{\text{cm}}}{\Delta\tau} < R$, the leading term is strictly reduced in the second sampling step. Furthermore, if $\epsilon_{\text{cm}} \approx \Delta\tau$, $W_2(\hat{P}_0^{(2)}, P_{\text{data}}) \approx \frac{2}{3}W_2(\hat{P}_0^{(1)}, P_{\text{data}})$; if $\epsilon_{\text{cm}} \ll \Delta\tau$, $W_2(\hat{P}_0^{(2)}, P_{\text{data}}) \approx \frac{1}{2}W_2(\hat{P}_0^{(1)}, P_{\text{data}})$. Due to the constraint in (10), further improvement with this heuristic is challenging. This intuition aligns with the empirical result in Luo et al. (2023).

**Case study 2:** In the second case study, we consider the following *Variance Exploding SDE* (Song et al., 2021; Karras et al., 2022):

$$\mathrm{d}\mathbf{x}_t = \sqrt{2t}\mathrm{d}\mathbf{w}_t, \quad (12)$$

which is used in Song et al. (2023) and Song & Dhariwal (2024) as the forward process for consistency models. The noise schedule is $(\alpha_t, \sigma_t^2) = (1, t^2)$ and the marginal distribution of $\mathbf{x}_t$ conditioning on $\mathbf{x}_0$ is: $\mathbf{x}_t \sim \mathcal{N}(\mathbf{x}_0, t^2I)$.The upper bound in (7) is simplified to:

$$W_2(\hat{P}_{t_N}, P_{\text{data}}) \leq 2R\underbrace{\left(\frac{1}{4t_1^2}R^2 + \sum_{j=2}^{N}\frac{1}{4t_j^2}t_{j-1}^2\frac{\epsilon_{\text{cm}}^2}{\Delta\tau^2}\right)^{1/4}}_{\text{(i)}} + \underbrace{t_N\frac{\epsilon_{\text{cm}}}{\Delta\tau}}_{\text{(ii)}}. \quad (13)$$

---

[5]When $\frac{\epsilon_{\text{cm}}}{\Delta\tau} \geq R$, $(9) = \Omega(R)$, which is meaningless because the support of $P_{\text{data}}$ is bounded by $R$ already. This trivial situation is not the focus of this case study.

This implies a trade-off in multi-step sampling with this particular forward process (12) when increasing the number of steps. Roughly speaking, (i) in (13) increases due to more terms with more steps while $t_N$ becomes smaller and (ii) will decrease. One consideration is to decrease $t_i$ by half in each sampling step until $t_N = \Delta\tau$ (ignore the rounding issue):

$$t_i = T2^{1-i}, \quad i = 1, 2, \ldots, \log_2\left(\frac{2T}{\Delta\tau}\right), \tag{14}$$

where $T > 0$ is to be determined. With this choice, (i) increases at a linear rate while (ii) decreases exponentially when using more sampling steps:

**Corollary 2** (Multistep sampling with the variance exploding SDE)**.** *Suppose the conditions in Theorem 1 are satisfied. Suppose $\alpha_t = 1$, $\sigma_t^2 = t^2$. Let $N = \log_2(2T)$. Then for $\{t_i\}_{i=1:N}$ defined in (14), we have: $W_2(\hat{P}_0^{(N)}, P_{data}) \leq O\left(R\sqrt{R}T^{-1/2} + R\sqrt{\frac{\epsilon_{cm}}{\Delta\tau}}\left(\log\frac{T}{\Delta\tau}\right)^{1/4}\right)$.*

When $T = \frac{R\Delta\tau}{\epsilon_{cm}}$, we have $W_2(\hat{P}_0^{(N)}, P_{\text{data}}) \leq O\left(R\sqrt{\frac{\epsilon_{cm}}{\Delta\tau}}\left(\log\frac{R}{\epsilon_{cm}}\right)^{1/4}\right)$. In this case study and the halving strategy for choosing sample time schedule $\{t_i\}_{i=1:N}$, reducing the partition size $\Delta\tau$ is beneficial only when the consistency loss $\epsilon_{cm}$ decreases at a faster rate.

In general, the convergence of a forward process in (1) is characterized by $\alpha_t^2\sigma_t^{-2}$ (according to Lemma 3). The forward process (12) has a polynomial convergence rate $\alpha_t^2\sigma_t^{2^{-2}} = t^{-2}$ while (8) enjoys a much faster exponential rate $\alpha_t^2\sigma_t^{-2} \approx e^{-2t}$. The exponential convergence results in a *shorter training step $T$*, *fewer sampling steps $N$*, and *better sample quality* if Assumption 1 holds with the same $\epsilon_{cm}$ in both cases.

## 4 TECHNICAL OVERVIEW

In this section, we present the high-level ideas in the proof for our main result Theorem 1 since proofs for Theorem 2 and 3 share the same main building blocks. The proof for Theorem 1 consists of three main components:

**Error decomposition:**   intuitively, the error comes from: (i) inaccurate consistency function $\hat{f}(\cdot, \cdot)$ and (ii) sampling from Gaussian distribution $\mathcal{N}(0, \sigma_{t_1}^2)$ instead of perturbed data distribution $P_{t_1}$. (i) is controlled by the consistency loss Assumption 1 and (ii) is controlled by the convergence of the forward process Lemma 3. However, the error (i) and (ii) interact with each other in the multi-step sampling. We handle this complication progressively, starting with the error decomposition in the final sampling step:

$$W_2(\hat{P}_0^{(N)}, P_{\text{data}}) \leq W_2(\hat{f}(\hat{P}_{t_N}, t_N), \hat{f}(P_{t_N}, t_N)) + W_2(\hat{f}(P_{t_N}, t_N), f^\star(P_{t_N}, t_N)).$$

Since the output of $\hat{f}(\cdot, \cdot)$ is *bounded*, we could simplify the first term with the TV distance, which is further upper bounded by $\text{KL}(P_{t_N} \| \hat{P}_{t_N})$ by *Pinsker's inequality* and *data processing inequality*. The second term is solely controlled by the consistency loss $\epsilon_{cm}$.

**Recursion on** $\text{KL}(P_{t_i} \| \hat{P}_{t_i})$**:**   we analyze $\text{KL}(P_{t_N} \| \hat{P}_{t_N})$ via *induction*. First of all, the base case $\text{KL}(P_{t_1} \| \hat{P}_{t_1})$ is upper bounded using the *convergence of the forward process*; the induction step connects $\text{KL}(P_{t_i} \| \hat{P}_{t_i})$ and $\text{KL}(P_{t_{i+1}} \| \hat{P}_{t_{i+1}})$. According to the multi-step sampling, $\hat{P}_{t_i}$ and $\hat{P}_{t_{i+1}}$ is connected by $\hat{f}(\cdot, t_i)$ and $\mathcal{D}\left(\cdot; \alpha_{t_{i+1}}, \sigma_{t_{i+1}}^2\right)$ as

$$\hat{P}_{t_i} \xrightarrow{\hat{f}(\cdot, t_i)} \hat{P}_0^{(i)} \xrightarrow{\mathcal{D}\left(\cdot; \alpha_{t_{i+1}}, \sigma_{t_{i+1}}^2\right)} \hat{P}_{t_{i+1}}.$$

In this process, $\hat{f}(\cdot, \cdot)$ induced additional error while the forward process $\mathcal{D}\left(\cdot; \alpha_{t_{i+1}}, \sigma_{t_{i+1}}^2\right)$ reduces it with convergence $\alpha_{t_{i+1}}^2\sigma_{t_{i+1}}^{-2}$. This intuition is formalized by the error decomposition via *chain rule of KL divergence*:

$$\text{KL}(P_{t_{i+1}} \| \hat{P}_{t_{i+1}}) \leq \text{KL}(P_{t_i} \| \hat{P}_{t_i}) + \frac{\alpha_{t_{i+1}}^2}{2\sigma_{t_{i+1}}^2}\mathbb{E}_{\mathbf{x} \sim P_{t_i}}\left[\left\|f^\star(\mathbf{x}, t_i) - \hat{f}(\mathbf{x}, t_i)\right\|_2^2\right].$$

Another possibility is to construct the recursive formula for $W_2(\hat{P}_0^{(i)}, P_0)$. However, recursion on $W_2$ requires the translation from KL to $W_2$ that induces an $R$ factor in each induction step. When $\{t_i\}_i$ is not carefully designed, the $R$ in each induction step results in an exploding upper bound easily. Meanwhile, this translation requires the data distribution to be bounded and hampers the application to more general data distributions.

**Error of consistency function evaluation:** another importance building block in our proof is the evaluation error of consistency function, i.e. $\left\|\hat{f}(\mathbf{x}, \tau_k) - f^\star(\mathbf{x}, \tau_k)\right\|_2$ for $\tau_k \in \mathcal{T}$. Assumption 1 controls the difference in $\hat{f}(\cdot, \cdot)$ and $f^\star(\cdot, \cdot)$ indirectly by enforcing the consistency property. We connect the evaluation error and consistency loss via a stepwise decomposition. Conditioning on $\mathbf{x}_{\tau_k} \sim P_{\tau_k}$, the PF-ODE (2) defines a deterministic trajectory:

$$\mathbf{x}_{\tau_k} \xrightarrow{\varphi(\tau_{k-1};\cdot,\tau_k)} \mathbf{x}_{\tau_{k-1}} \xrightarrow{\varphi(\tau_{k-2};\cdot,\tau_{k-1})} \mathbf{x}_{\tau_{k-2}} \xrightarrow{\varphi(\tau_{k-3};\cdot,\tau_{k-2})} \cdots \xrightarrow{\varphi(\tau_1;\cdot,\tau_2)} \mathbf{x}_{\tau_1} \xrightarrow{\varphi(\tau_0;\cdot,\tau_1)} \mathbf{x}_{\tau_0}.$$

Assumption 1 guarantees that $\left\|\hat{f}(\mathbf{x}_{\tau_j}, \tau_j) - \hat{f}(\mathbf{x}_{\tau_{j-1}}, \tau_{j-1})\right\|_2$ is small in the sense of $L_2$ error for each intermediate step $j$. We could make the following decomposition:

$$\left\|\hat{f}(\mathbf{x}_{\tau_k}, \tau_k) - f^\star(\mathbf{x}_{\tau_k}, \tau_k)\right\|_2 = \left\|\hat{f}(\mathbf{x}_{\tau_k}, \tau_k) - \mathbf{x}_0\right\|_2 \leq \sum_{j=1}^{k} \left\|\hat{f}(\mathbf{x}_{\tau_j}, \tau_j) - \hat{f}(\mathbf{x}_{\tau_{j-1}}, \tau_{j-1})\right\|_2$$

The right-hand side is, roughly speaking $\leq \tau_k \frac{\epsilon_{\mathrm{cm}}}{\Delta\tau}$, We formalize this idea with *Minkowski inequality* in Lemma 2.

## 5 CONCLUSION

In this paper, we study the convergence of the consistency model multistep sampling procedure. We establish guarantees on the distance between the sample distribution and data distribution in terms of both Wasserstein distance and total variation distribution. Our upper bound requires only mild assumptions on the data distribution.

Future research directions include providing lower bounds on multistep sampling and establishing end-to-end results on consistency models.

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

## A  MULTISTEP SAMPLING

We present the multistep sampling procedure in Algorithm 1. Compared to Algorithm 1 of Song et al. (2023), we allow different choices of noise schedule in Algorithm 1.

---

**Algorithm 1** Multistep Consistency Sampling

1: **Input:** a trained consistency model $\hat{f}(\cdot,\cdot)$, noise schedule $\left\{(\alpha_t, \sigma_t^2)\right\}_{t\in[0,T]}$, sampling time schedule $\{t_i\}_{i=1:N}$, where $t_N = T$.
2: $\hat{\mathbf{x}}_{t_1}^{(1)} \sim \mathcal{N}(0, \sigma_{t_1}^2 I)$
3: **for** $i = 1$ to $N-1$ **do**
4: $\quad \hat{\mathbf{x}}_0^{(i)} \leftarrow \hat{f}(\hat{\mathbf{x}}_{t_i}^{(i)}, t_i)$
5: $\quad \hat{\mathbf{x}}_{t_{i+1}}^{(i+1)} \sim \mathcal{N}(\alpha_{t_{i+1}}\hat{\mathbf{x}}_0^{(i)}, \sigma_{t_{i+1}}^2 I)$
6: **end for**
7: **Output:** $\hat{\mathbf{x}}_0^{(N)}$.

---

## B  PROOF OF THEOREM 1

At a high level, we could decompose the $W_2$ error $W_2(\hat{P}_0^{(N)}, P_{\text{data}})$ into:

$$W_2(\hat{P}_0^{(N)}, P_{\text{data}}) \leq W_2(\hat{P}_0^{(N)}, \hat{f}(P_{t_N}, t_N)) + W_2(\hat{f}(P_{t_N}, t_N), P_{\text{data}})$$
$$= \underbrace{W_2(\hat{f}(\hat{P}_{t_N}, t_N), \hat{f}(P_{t_N}, t_N))}_{=:\mathscr{A}_1} + \underbrace{W_2(\hat{f}(P_{t_N}, t_N), f^\star(P_{t_N}, t_N))}_{=:\mathscr{A}_2}. \quad (15)$$

In the error decomposition (15): the first term $\mathscr{A}_1$ is caused by an inaccurate noise distribution $\hat{P}_{t_N}$ and is controlled by the KL divergence of $P_{t_N}$ from $\hat{P}_{t_N}$. We use the chain rule of KL divergence to derive a recursive formula for $\text{KL}(P_{t_i} \parallel \hat{P}_{t_i})$, where the initial term $\text{KL}(P_{t_1} \parallel \hat{P}_{t_1})$ is bounded by the convergence of the forward diffusion process:

**Lemma 1** (Decomposition of KL). *Suppose $\hat{f}(\cdot,\cdot)$ satisfies Assumption 1, then for all $i = 1, \ldots, N$, we have:*

$$\text{KL}(P_{t_i} \parallel \hat{P}_{t_i}) \leq \frac{\alpha_{t_1}^2}{2\sigma_{t_1}^2} \mathbb{E}_{\mathbf{x}\sim P_{data}}\left[\|\mathbf{x}\|_2^2\right] + \sum_{j=2}^{i} \frac{\alpha_{t_j}^2}{2\sigma_{t_j}^2} t_{j-1}^2 \frac{\epsilon_{\text{cm}}^2}{\Delta\tau^2}.$$

We defer the proof of Lemma 1 to Section B.1. Given this result, we can bound $\mathscr{A}_1$ as:

$$\mathscr{A}_1 \leq 2R\sqrt{\text{TV}(\hat{f}(\hat{P}_{t_N}, t_N), \hat{f}(P_{t_N}, t_N))} \quad \left(\text{By Section 2.2.4 of Rolland (2022) and } \left\|\hat{f}(\mathbf{x}, t)\right\|_2 \leq R\right)$$

$$\leq 2R\left(\frac{1}{2}\text{KL}(\hat{f}(P_{t_N}, t_N) \parallel \hat{f}(\hat{P}_{t_N}, t_N))\right)^{1/4} \quad \text{(By Pinsker's inequality)}$$

$$\leq 2R\left(\frac{1}{2}\text{KL}(P_{t_N} \parallel \hat{P}_{t_N})\right)^{1/4} \quad \text{(By data processing inequality)}$$

$$\leq 2R \left( \frac{\alpha_{t_1}^2}{4\sigma_{t_1}^2} \mathbb{E}_{\mathbf{x} \sim P_{\text{data}}} \left[ \|\mathbf{x}\|_2^2 \right] + \sum_{j=2}^{N} \frac{\alpha_{t_j}^2}{4\sigma_{t_j}^2} t_{j-1}^2 \frac{\epsilon_{\text{cm}}^2}{\Delta\tau^2} \right)^{1/4} \qquad \text{(By Lemma 1 with } i = N)$$

$$\leq 2R \left( \frac{\alpha_{t_1}^2}{4\sigma_{t_1}^2} R^2 + \sum_{j=2}^{N} \frac{\alpha_{t_j}^2}{4\sigma_{t_j}^2} t_{j-1}^2 \frac{\epsilon_{\text{cm}}^2}{\Delta\tau^2} \right)^{1/4} \qquad \left( \text{Because } \sup_{\mathbf{x} \in \text{supp}(P_{\text{data}})} \|\mathbf{x}\|_2 \leq R \right). \tag{16}$$

The second term $\mathscr{A}_2$ is caused by the difference between the pre-trained consistency function $\hat{f}(\cdot, \cdot)$ and the ground truth $f^\star(\cdot, \cdot)$, which is controlled by the consistency loss $\epsilon_{\text{cm}}$.

**Lemma 2.** *Suppose $\hat{f}(\cdot, \cdot)$ satisfies Assumption 1 holds, then for all $i = 0, 1, \ldots, M$, we have:*

(i) $\mathbb{E}_{\mathbf{x} \sim P_{\tau_i}} \left[ \left\| \hat{f}(\mathbf{x}, \tau_i) - f^\star(\mathbf{x}, \tau_i) \right\|_2^2 \right] \leq \tau_i^2 \frac{\epsilon_{\text{cm}}^2}{\Delta\tau^2};$

(ii) $W_2(\hat{f}(P_{\tau_i}, \tau_i), f^\star(P_{\tau_i}, \tau_i)) \leq \tau_i \frac{\epsilon_{\text{cm}}}{\Delta\tau}.$

We defer the proof of Lemma 2 to Section B.1. Part (ii) of Lemma 2 shows that:

$$\mathscr{A}_2 \leq t_N \frac{\epsilon_{\text{cm}}}{\Delta\tau}. \tag{17}$$

We finish the proof of Theorem 1 by combining (16) and (17).

### B.1 PROOF OF AUXILIARY LEMMAS

*Proof of Lemma 1.* We prove this statement via induction. At a high level, the base is proved by the convergence of the forward process Lemma 3. We show the induction step by the chain rule of KL.

When $i = 1$, we have can write $\hat{P}_{t_1} = \mathcal{N}(0, \sigma_{t_1}^2)$ with the diffusion operator and a the dirac distribution:

$$\hat{P}_{t_1} = \mathcal{D}\left( \delta_0; \alpha_{t_1}, \sigma_{t_1}^2 \right),$$

where $\delta_0$ is the delta distribution at $0$. By definition, $P_{t_1} = \mathcal{D}\left( P_0; \alpha_{t_1}, \sigma_{t_1}^2 \right)$. By Lemma 3,

$$\text{KL}(P_{t_1} \| \hat{P}_{t_1}) = \text{KL}(\mathcal{D}\left( P_0; \alpha_{t_1}, \sigma_{t_1}^2 \right) \| \mathcal{D}\left( \delta_0; \alpha_{t_1}, \sigma_{t_1}^2 \right))$$

$$\leq \frac{\alpha_{t_1}^2}{2\sigma_{t_1}^2} W_2^2(P_0, \delta_0) = \frac{\alpha_{t_1}^2}{2\sigma_{t_1}^2} \mathbb{E}_{\mathbf{x} \in P_{\text{data}}} \left[ \|\mathbf{x}\|_2^2 \right].$$

Thus the statement holds for $i = 1$. Suppose the statement holds for $i = k$, i.e.

$$\text{KL}(P_{t_k} \| \hat{P}_{t_k}) \leq \frac{\alpha_{t_1}^2}{2\sigma_{t_1}^2} \mathbb{E}_{\mathbf{x} \sim P_{\text{data}}} \left[ \|\mathbf{x}\|_2^2 \right] + \sum_{j=2}^{k} \frac{\alpha_{t_j}^2}{2\sigma_{t_j}^2} t_{j-1}^2 \frac{\epsilon_{\text{cm}}^2}{\Delta\tau^2}. \tag{18}$$

We first explicitly write the sequence of random variables in the multistep inference:

$$\hat{\mathbf{x}}_{t_1}^{(1)} \to \hat{\mathbf{x}}_0^{(1)} \to \hat{\mathbf{x}}_{t_2}^{(2)} \to \hat{\mathbf{x}}_0^{(2)} \to \cdots \to \hat{\mathbf{x}}_{t_N}^{(N)} \to \hat{\mathbf{x}}_0^{(N)},$$

where $\hat{\mathbf{x}}_{t_1}^{(1)} \sim \mathcal{N}(0, \sigma_{t_1}^2 I)$, $\hat{\mathbf{x}}_0^{(i)} = \hat{f}(\hat{\mathbf{x}}_0^{(i)}, t_i)$, $\hat{\mathbf{x}}_{t_{i+1}}^{(i+1)} \sim \mathcal{N}(\alpha_{t_{i+1}} \hat{\mathbf{x}}_0^{(i+1)}, \sigma_{t_{i+1}}^2 I)$. Similarly, we also define the following process that starts at the ground truth noise distribution $P_{t_1}$ and evolves using the ground truth consistency function $f^\star(\cdot, \cdot)$ :

$$\mathbf{x}_{t_1}^{(1)} \to \mathbf{x}_0^{(1)} \to \mathbf{x}_{t_2}^{(2)} \to \mathbf{x}_0^{(2)} \to \cdots \to \mathbf{x}_{t_N}^{(N)} \to \mathbf{x}_0^{(N)},$$

where $\mathbf{x}_{t_1}^{(1)} \sim P_{t_1}$, $\mathbf{x}_0^{(i)} = f^\star(\mathbf{x}_0^{(i)}, t_i)$, $\mathbf{x}_{t_{i+1}}^{(i+1)} \sim \mathcal{N}(\alpha_{t_{i+1}} \mathbf{x}_0^{(i)}, \sigma_{t_{i+1}}^2 I)$.

By the chain rule of KL divergence, we have:

$$\text{KL}(\mathcal{P}\left( \mathbf{x}_{t_{k+1}}^{(k+1)} \right) \| \mathcal{P}\left( \hat{\mathbf{x}}_{t_{k+1}}^{(k+1)} \right))$$

$$+ \underbrace{\mathbb{E}_{\mathbf{x} \sim \mathcal{P}\left(\mathbf{x}_{t_{k+1}}^{(k+1)}\right)} \left[ \mathrm{KL}(\mathcal{P}\left(\mathbf{x}_{t_k}^{(k)} | \mathbf{x}_{t_{k+1}}^{(k+1)} = \mathbf{x}\right) \parallel \mathcal{P}\left(\hat{\mathbf{x}}_{t_k}^{(k)} | \hat{\mathbf{x}}_{t_{k+1}}^{(k+1)} = \mathbf{x}\right)) \right]}_{\geq 0}$$

$$= \mathrm{KL}(\mathcal{P}\left(\mathbf{x}_{t_k}^{(k)}, \mathbf{x}_{t_{k+1}}^{(k+1)}\right) \parallel \mathcal{P}\left(\hat{\mathbf{x}}_{t_k}^{(k)}, \hat{\mathbf{x}}_{t_{k+1}}^{(k+1)}\right))$$

$$= \mathrm{KL}(\mathcal{P}\left(\mathbf{x}_{t_k}^{(k)}\right) \parallel \mathcal{P}\left(\hat{\mathbf{x}}_{t_k}^{(k)}\right)) + \mathbb{E}_{\mathbf{x} \sim \mathcal{P}\left(\mathbf{x}_{t_k}^{(k)}\right)} \left[ \mathrm{KL}(\mathcal{P}\left(\mathbf{x}_{t_{k+1}}^{(k+1)} | \mathbf{x}_{t_k}^{(k)} = \mathbf{x}\right) \parallel \mathcal{P}\left(\hat{\mathbf{x}}_{t_{k+1}}^{(k+1)} | \hat{\mathbf{x}}_{t_k}^{(k)} = \mathbf{x}\right)) \right]$$

where we use $\mathcal{P}(\mathbf{x})$ to denote the distribution of random variable $\mathbf{x}$. Because KL is non-negative, we have:

$$\mathrm{KL}(\mathcal{P}\left(\mathbf{x}_{t_{k+1}}^{(k+1)}\right) \parallel \mathcal{P}\left(\hat{\mathbf{x}}_{t_{k+1}}^{(k+1)}\right))$$

$$\leq \mathrm{KL}(\mathcal{P}\left(\mathbf{x}_{t_k}^{(k)}\right) \parallel \mathcal{P}\left(\hat{\mathbf{x}}_{t_k}^{(k)}\right)) + \mathbb{E}_{\mathbf{x} \sim \mathcal{P}\left(\mathbf{x}_{t_k}^{(k)}\right)} \left[ \mathrm{KL}(\mathcal{P}\left(\mathbf{x}_{t_{k+1}}^{(k+1)} | \mathbf{x}_{t_k}^{(k)} = \mathbf{x}\right) \parallel \mathcal{P}\left(\hat{\mathbf{x}}_{t_{k+1}}^{(k+1)} | \hat{\mathbf{x}}_{t_k}^{(k)} = \mathbf{x}\right)) \right]$$

By definition, this means:

$$\mathrm{KL}(P_{t_{k+1}} \parallel \hat{P}_{t_{k+1}})$$

$$\leq \mathrm{KL}(P_{t_k} \parallel \hat{P}_{t_k}) + \mathbb{E}_{\mathbf{x} \sim P_{t_k}} \left[ \mathrm{KL}(\mathcal{N}(\alpha_{t_{k+1}} f^{\star}(\mathbf{x}, t_k), \sigma_{t_{k+1}}^2 I) \parallel \mathcal{N}(\alpha_{t_{k+1}} \hat{f}(\mathbf{x}, t_k), \sigma_{t_{k+1}}^2 I)) \right]$$

$$= \mathrm{KL}(P_{t_k} \parallel \hat{P}_{t_k}) + \frac{\alpha_{t_{k+1}}^2}{2\sigma_{t_{k+1}}^2} \mathbb{E}_{\mathbf{x} \sim P_{t_k}} \left[ \left\| f^{\star}(\mathbf{x}, t_k) - \hat{f}(\mathbf{x}, t_k) \right\|_2^2 \right]$$

$$\leq \mathrm{KL}(P_{t_k} \parallel \hat{P}_{t_k}) + \frac{\alpha_{t_{k+1}}^2}{2\sigma_{t_{k+1}}^2} t_k^2 \frac{\epsilon_{\mathrm{cm}}^2}{\Delta \tau^2} \quad \text{(By part (i) of Lemma 2)}$$

$$\leq \frac{\alpha_{t_1}^2}{2\sigma_{t_1}^2} \mathbb{E}_{\mathbf{x} \sim P_{\mathrm{data}}} \left[ \|\mathbf{x}\|_2^2 \right] + \sum_{j=2}^{k+1} \frac{\alpha_{t_j}^2}{2\sigma_{t_j}^2} t_{j-1}^2 \frac{\epsilon_{\mathrm{cm}}^2}{\Delta \tau^2}. \quad \text{(By (18))}$$

$$\square$$

*Proof of Lemma 2.* We first prove part (i) with induction on $t$. By the definition of $f^{\star}(\cdot, \cdot)$ in (3),

$$f^{\star}(\mathbf{x}, 0) = \varphi(0; \mathbf{x}, 0) = \mathbf{x}, \quad \forall \mathbf{x} \in \mathbb{R}^d.$$

By Assumption 1, $\hat{f}(\mathbf{x}, 0) = \mathbf{x}$ for all $\mathbf{x}$. Thus

$$\mathbb{E}_{\mathbf{x} \sim P_0} \left[ \left\| \hat{f}(\mathbf{x}, 0) - f^{\star}(\mathbf{x}, 0) \right\|_2^2 \right] = \mathbb{E}_{\mathbf{x} \sim P_0} \left[ \|\mathbf{x} - \mathbf{x}\|_2^2 \right] = 0,$$

which means (i) holds for $i = 0$.

Suppose (i) holds for $i = s$, i.e.

$$\sqrt{\mathbb{E}_{\mathbf{x} \sim P_{\tau_s}} \left[ \left\| \hat{f}(\mathbf{x}, \tau_s) - f^{\star}(\mathbf{x}, \tau_s) \right\|_2^2 \right]} \leq \tau_s \epsilon_{\mathrm{cm}} / \Delta \tau. \quad (19)$$

By the property of the PF-ODE (2),

$$\varphi(\tau_{s+1}; \mathbf{x}, \tau_s) \sim P_{\tau_{s+1}}, \quad \text{if } \mathbf{x} \sim P_{\tau_s}. \quad (20)$$

When $i = s + 1$, we have:

$$\sqrt{\mathbb{E}_{\mathbf{x}' \sim P_{\tau_{s+1}}} \left[ \left\| \hat{f}(\mathbf{x}', \tau_{s+1}) - f^{\star}(\mathbf{x}', \tau_{s+1}) \right\|_2^2 \right]}$$

$$= \sqrt{\mathbb{E}_{\mathbf{x} \sim P_{\tau_s}} \left[ \left\| \hat{f}(\varphi(\tau_{s+1}; \mathbf{x}, \tau_s), \tau_{s+1}) - f^{\star}(\varphi(\tau_{s+1}; \mathbf{x}, \tau_s), \tau_{s+1}) \right\|_2^2 \right]} \quad \text{(By (20))}$$

$$= \sqrt{\mathbb{E}_{\mathbf{x} \sim P_{\tau_s}} \left[ \left\| \hat{f}(\varphi(\tau_{s+1}; \mathbf{x}, \tau_s), \tau_{s+1}) - f^{\star}(\mathbf{x}, \tau_s) \right\|_2^2 \right]} \quad \text{(By the definition of } f^{\star}(\cdot, \cdot))$$

$$\leq \sqrt{\mathbb{E}_{\mathbf{x}\sim P_{\tau_s}}\left[\left\|\hat{f}(\varphi(\tau_{s+1};\mathbf{x},\tau_s),\tau_{s+1}) - \hat{f}(\mathbf{x},\tau_s)\right\|_2^2\right]} + \sqrt{\mathbb{E}_{\mathbf{x}\sim P_{\tau_s}}\left[\left\|\hat{f}(\mathbf{x},\tau_s) - f^\star(\mathbf{x},\tau_s)\right\|_2^2\right]}$$

(By Lemma 5)

$$\leq \epsilon_{\text{cm}} + \tau_s\epsilon_{\text{cm}}/\Delta\tau \quad \text{(By Assumption 1 and (19))}$$

$$= \epsilon_{\text{cm}}(1 + \tau_s/\Delta\tau) = \tau_{s+1}\epsilon_{\text{cm}}/\Delta\tau.$$

We complete the proof for part (i).

$\hat{f}(\cdot,t)$ and $f^\star(\cdot,t)$ induce a joint distribution $\Gamma_{\mathbf{x}_0',\mathbf{x}_0}$:

$$\text{Pr}_{(\mathbf{x}_0',\mathbf{x}_0)\sim\Gamma_{\mathbf{x}_0',\mathbf{x}_0}}[(\mathbf{x}_0',\mathbf{x}_0)\in A] := \text{Pr}_{\mathbf{x}_t\sim P_t}\left[\mathbf{x}_t \in \left\{\mathbf{x}:(\hat{f}(\mathbf{x},t),f^\star(\mathbf{x},t))\in A\right\}\right],$$

for any event $A$. With this joint distribution $\Gamma_{\mathbf{x}_0',\mathbf{x}_0}$, the marginal distribution of $\mathbf{x}_0'$ is $\hat{f}(P_t,t)$ and the marginal distribution of $\mathbf{x}_0$ is $f^\star(P_t,t)$. This means:

$$\sqrt{\mathbb{E}_{\mathbf{x}_t\sim P_t}\left[\left\|\hat{f}(\mathbf{x}_t,t) - f^\star(\mathbf{x}_t,t)\right\|_2^2\right]} = \sqrt{\mathbb{E}_{(\mathbf{x}_0',\mathbf{x}_0)\sim\Gamma_{\mathbf{x}_0',\mathbf{x}_0}}\left[\|\mathbf{x}_0'-\mathbf{x}_0\|_2^2\right]} \geq W_2(\hat{f}(P_t,t),f^\star(P_t,t)).$$

By applying part (i), we have

$$W_2(\hat{f}(P_{\tau_i},\tau_i),f^\star(P_{\tau_i},\tau_i)) \leq \tau_i\epsilon_{\text{cm}}/\Delta\tau.$$

We complete the proof for part (ii). $\qquad\square$

## C  PROOF OF THEOREM 2

The error term can be decomposed as:

$$W_2(\hat{P}_0^{(t_N)}, P_{\text{data}}) \leq W_2(\hat{P}_0^{(t_N)}, P_{\text{data}\cap\mathcal{B}(0,R)}) + W_2(P_{\text{data}\cap\mathcal{B}(0,R)}, P_{\text{data}}) \tag{21}$$

By Theorem 1,

$$W_2(\hat{P}_0^{(t_N)}, P_{\text{data}\cap\mathcal{B}(0,R)}) \leq 2R\left(\frac{\alpha_{t_1}^2}{4\sigma_{t_1}^2}R^2 + \sum_{j=2}^N \frac{\alpha_{t_j}^2}{4\sigma_{t_j}^2}t_{j-1}^2\epsilon_{\text{cm}}^2\right)^{1/4} + t_N\epsilon_{\text{cm}}.$$

For the second term, we first note that

$$\text{TV}(P_{\text{data}\cap\mathcal{B}(0,R)}, P_{\text{data}}) = \text{Pr}_{\mathbf{x}\sim P_{\text{data}}}[\|\mathbf{x}\|_2 > R] \leq O(e^{-\frac{R}{C}}).$$

By Lemma 9 of Rolland (2022),

$$W_2(P_{\text{data}\cap\mathcal{B}(0,R)}, P_{\text{data}}) \leq O(Re^{-\frac{R}{2C}}).$$

We finish the proof by combining these two bounds.

## D  PROOF OF THEOREM 3

At a high level, we can decompose the TV distance as follows:

$$\text{TV}(\hat{P}_0^{(N)} * \mathcal{N}(0,\sigma_\epsilon^2 I), P_{\text{data}})$$

$$\leq \text{TV}(\hat{P}_0^{(N)} * \mathcal{N}(0,\sigma_\epsilon^2 I), P_{\text{data}} * \mathcal{N}(0,\sigma_\epsilon^2 I)) + \text{TV}(P_{\text{data}} * \mathcal{N}(0,\sigma_\epsilon^2 I), P_{\text{data}}) \tag{22}$$

The first term can be bounded by Lemma 1 and Pinsker's inequality, which shows that the TV distance between $\hat{P}_0^{(N)}$ and $P_{\text{data}}$ is controlled after the Gaussian perturbation. While the second term is bounded when $P_{\text{data}}$ satisfies the smoothness assumption, which shows that the perturbation will change $P_{\text{data}}$ only slightly. We now illustrate these ideas in detail. We first define $\alpha_{t_{N+1}} := 1$, $\sigma_{t_{N+1}} := \sigma_\epsilon$, then by Pinsker's inequality and Lemma 1:

$$\text{TV}(\hat{P}_0^{(N)} * \mathcal{N}(0,\sigma_\epsilon^2 I), P_{\text{data}} * \mathcal{N}(0,\sigma_\epsilon^2 I))$$

$$\leq \sqrt{\frac{1}{2} \operatorname{KL}(P_{\text{data}} * \mathcal{N}(0, \sigma_\epsilon^2 I) \parallel \hat{P}_0^{(N)} * \mathcal{N}(0, \sigma_\epsilon^2 I))}$$

$$= \sqrt{\frac{1}{2} \operatorname{KL}(P_{t_{N+1}} \parallel \hat{P}_{t_{N+1}})}$$

$$\leq \sqrt{\frac{\alpha_{t_1}^2}{4\sigma_{t_1}^2} \mathbb{E}_{\mathbf{x} \sim P_{\text{data}}} \left[\|\mathbf{x}\|_2^2\right] + \sum_{j=2}^{N+1} \frac{\alpha_{t_j}^2}{4\sigma_{t_j}^2} t_{j-1}^2 \epsilon_{\text{cm}}^2}$$

$$= \sqrt{\frac{\alpha_{t_1}^2}{4\sigma_{t_1}^2} \mathbb{E}_{\mathbf{x} \sim P_{\text{data}}} \left[\|\mathbf{x}\|_2^2\right] + \sum_{j=2}^{N} \frac{\alpha_{t_j}^2}{4\sigma_{t_j}^2} t_{j-1}^2 \epsilon_{\text{cm}}^2 + \frac{1}{4\sigma_\epsilon^2} t_N^2 \epsilon_{\text{cm}}^2}$$

$$\leq \sqrt{\frac{\alpha_{t_1}^2}{4\sigma_{t_1}^2} \mathbb{E}_{\mathbf{x} \sim P_{\text{data}}} \left[\|\mathbf{x}\|_2^2\right] + \sum_{j=2}^{N} \frac{\alpha_{t_j}^2}{4\sigma_{t_j}^2} t_{j-1}^2 \epsilon_{\text{cm}}^2 + \frac{1}{2\sigma_\epsilon} t_N \epsilon_{\text{cm}}}.$$

On the other hand, by Lemma 4,

$$\operatorname{TV}(P_{\text{data}} * \mathcal{N}(0, \sigma_\epsilon^2 I), P_{\text{data}}) \leq 2dL\sigma_\epsilon.$$

We complete the proof by combining these two bounds into the decomposition in (22).

# E  CONNECTION TO CONSISTENCY DISTILLATION

Our Assumption1 assumes that the self-consistency property is satisfied approximately, which aligns with both consistency distillation (Song et al., 2023). For simplicity, we consider an OU process to be the forward process:

$$\mathrm{d}\mathbf{x}_t = -\mathbf{x}_t \mathrm{d}t + \sqrt{2}\mathrm{d}\mathbf{w}_t, \quad \mathbf{x}_0 \sim P_{\text{data}}.$$

Given the pre-trained score function $s(\mathbf{x}, t)$, we train a consistency model from the following ODE:

$$\frac{\mathrm{d}\mathbf{x}_t}{\mathrm{d}t} = -\mathbf{x}_t - s(\mathbf{x}_t, t), \quad \mathbf{x}_T \sim \mathcal{N}(0, (1 - e^{-2T})I). \tag{23}$$

We assume access to an ODE solver, which can calculate $\varphi^s$, the solution to (23), exactly. Even though this solver can be computationally expensive during the training procedure, the consistency model will still be computationally efficient during the inference time.

To avoid distribution shift, we optimize the consistency loss objective (4) using the data generated from (23), instead of that from $P_t$, the marginal distribution of the forward process. When optimized properly, we can find a $\hat{f}$, s.t.

$$\mathbb{E}_{\mathbf{x}_{\tau_i} \sim \varphi^s(\tau_i; \mathcal{N}(0, (1-e^{-2T})I), T)} \left[\left\|\hat{f}(\mathbf{x}_{\tau_i}, \tau_i) - \hat{f}(\varphi(\tau_{i+1}; \mathbf{x}_{\tau_i}, \tau_i), \tau_{i+1})\right\|_2^2\right] \tag{24}$$

is small for all $i$. Using the same argument in Lemma 4, we can show that $\hat{f}(\mathcal{N}(0, (1 - e^{-2T})I), T)$ and $\varphi^s(0; \mathcal{N}(0, (1 - e^{-2T})I), T)$ are close in $W_2$, this can be translated into a bound in TV using the argument in Section 3.2.

When the pre-trained score function $s(\mathbf{x}, t)$ has small $L_2$ error, Huang et al. (2024) show that $\varphi^s(0; \mathcal{N}(0, (1 - e^{-2T})I), T)$ is close to $P_{\text{data}}$ in TV. To conclude, $\hat{f}(\mathcal{N}(0, (1 - e^{-2T})I), T)$ is close to $P_{\text{data}}$ in TV.

# F  TECHNICAL LEMMAS

We first present the result on the convergence of SDE, which also connects KL-divergence and $W_2$:

**Lemma 3.** *Let $P$ and $Q$ be two distributions in $\mathbb{R}^d$, then*

$$\operatorname{KL}(\mathcal{D}\left(P; \alpha, \sigma^2\right) \parallel \mathcal{D}\left(Q; \alpha, \sigma^2\right)) \leq \frac{\alpha^2}{2\sigma^2} W_2^2(P, Q)$$

This result is comparable to Lemma C.4 of Chen et al. (2023a). However, our results is self-contained and tighter.

*Proof of Lemma 3.* Let $U$ and $V$ be two random variables with joint distribution $\Gamma$, s.t. the marginal distributions of $U$ and $V$ are $P$ and $Q$ respectively. Let $X \sim \mathcal{D}\left(P; \alpha, \sigma^2\right)$ and $Y \sim \mathcal{D}\left(Q; \alpha, \sigma^2\right)$. We use $\mathcal{P}(\cdot)$ to denote the distribution of a random variable. By the chain rule of KL-divergence, we have:

$$\mathrm{KL}(\mathcal{P}(X) \parallel \mathcal{P}(Y)) \leq \mathrm{KL}(\mathcal{P}(X) \parallel \mathcal{P}(Y)) + \mathbb{E}_{\mathbf{x} \sim \mathcal{P}(X)}[\mathrm{KL}(\mathcal{P}((U,V)|X = \mathbf{x}) \parallel (U,V)|Y = \mathbf{x})]$$
$$\text{(By the non-negativity of KL)}$$
$$= \mathrm{KL}(\mathcal{P}(U,V) \parallel \mathcal{P}(U,V))$$
$$+ \mathbb{E}_{(\mathbf{u},\mathbf{x}) \sim \mathcal{P}(U,V)}[\mathrm{KL}(\mathcal{P}(X|(U,V) = (\mathbf{u},\mathbf{v}))) \parallel \mathcal{P}(Y|(U,V) = (\mathbf{u},vb)))]$$
$$\text{(By the chain rule of KL)}$$
$$= \mathbb{E}_{(\mathbf{u},\mathbf{x}) \sim \mathcal{P}(U,V)}[\mathrm{KL}(\mathcal{P}(X|U = \mathbf{u})) \parallel \mathcal{P}(Y|V = \mathbf{v}))] \tag{25}$$
$$(X \text{ is independent of } V \text{ given } U \text{ and similar holds for } Y)$$

By the definition of $\mathcal{D}\left(\cdot; \cdot, \cdot\right)$, $X|U = \mathbf{u} \sim \mathcal{N}(\alpha\mathbf{u}, \sigma^2 I)$ and $Y|V = \mathbf{v} \sim \mathcal{N}(\alpha\mathbf{v}, \sigma^2 I)$. Thus,

$$\mathrm{KL}(\mathcal{P}(X|U = \mathbf{u})) \parallel \mathcal{P}(Y|V = \mathbf{v})) = \frac{1}{2\sigma^2}\alpha^2 \|\mathbf{u} - \mathbf{v}\|_2^2$$

By (25), we further have:

$$\mathrm{KL}(\mathcal{D}\left(P; \alpha, \sigma^2\right) \parallel \mathcal{D}\left(Q; \alpha, \sigma^2\right)) \leq \frac{\alpha^2}{2\sigma^2}\mathbb{E}_{(\mathbf{u},\mathbf{v}) \sim \Gamma}\left[\|\mathbf{u} - \mathbf{v}\|_2^2\right] \tag{26}$$

By taking $\inf$ over $\Gamma$ on both sides of (26), we get:

$$\mathrm{KL}(\mathcal{D}\left(P; \alpha, \sigma^2\right) \parallel \mathcal{D}\left(Q; \alpha, \sigma^2\right)) \leq \frac{\alpha^2}{2\sigma^2}W_2^2(P, Q).$$

$\square$

**Lemma 4** (Gaussian perturbation on a smooth distribution, a variant of Lemma 6.4 of Lee et al. (2023)). *Let $P$ be a distribution in $\mathbb{R}^d$ with PDF $p(\mathbf{x})$, if $\log p(\mathbf{x})$ is L-smooth, then*
$$\mathrm{TV}(P, P * \mathcal{N}(0, \sigma^2 I)) \leq 2dL\sigma,$$
*where we use $P * Q$ to denote the convolution of distribution $P$ and $Q$.*

*Proof.* The results follows directly from Lemma 6.4 of Lee et al. (2023) with $\alpha_t = 1$ and $\sigma_t = \sigma$. $\square$

**Lemma 5** (Triangle inequality with both $L_p$ norm and $L_2$ norm). *Let $\mathbf{x}$ be a random variable in $\mathbb{R}^d$, and $f, g$ be mappings from $\mathbb{R}^d$ to $\mathbb{R}^d$, then*
$$\mathbb{E}_{\mathbf{x}}[\|f(\mathbf{x}) + g(\mathbf{x})\|_2^p]^{1/p} \leq \mathbb{E}_{\mathbf{x}}[\|f(\mathbf{x})\|_2^p]^{1/p} + \mathbb{E}_{\mathbf{x}}[\|g(\mathbf{x})\|_2^p]^{1/p}.$$

*Proof.*
$$\mathbb{E}_{\mathbf{x}}[\|f(\mathbf{x}) + g(\mathbf{x})\|_2^p]^{1/p} \leq \mathbb{E}_{\mathbf{x}}[(\|f(\mathbf{x})\|_2 + \|g(\mathbf{x})\|_2)^p]^{1/p} \quad \text{(Triangle inequality for } L_2 \text{ norm)}$$
$$\leq \mathbb{E}_{\mathbf{x}}[\|f(\mathbf{x})\|_2^p]^{1/p} + \mathbb{E}_{\mathbf{x}}[\|g(\mathbf{x})\|_2^p]^{1/p} \quad \text{(Minkowski inequality)}.$$

$\square$

## G SIMULATION

NEW

**Motivations:** Consistency model has already demonstrated its power on large-scale image generation tasks (Luo et al., 2023; Song et al., 2023; Song & Dhariwal, 2024). To verify our theoretical findings, we focus on a toy example that is easier to interpret.

We first refine our upper bound in Theorem 1, where we relax our result for a cleaner presentation. We make adjustment to (16) and get:

$$\sup_{\mathbf{x},\mathbf{y} \in \mathrm{supp}(P_{\mathrm{data}})} \|\mathbf{x} - \mathbf{y}\|_2 \left(\frac{\alpha_{t_1}^2}{2\sigma_{t_1}^2}\mathbb{E}_{\mathbf{x} \sim P_{\mathrm{data}}}\left[\|\mathbf{x}\|_2^2\right] + \sum_{j=2}^{N}\frac{\alpha_{t_j}^2}{4\sigma_{t_j}^2}t_{j-1}^2\frac{\epsilon_{\mathrm{cm}}^2}{\Delta\tau^2}\right)^{1/4} + t_N\frac{\epsilon_{\mathrm{cm}}}{\Delta\tau}. \tag{27}$$

**Simulation setting:** We consider OU process as the forward process, which is our setup in **Case study 1**. For simplicity, we consider a Bernoulli data distribution: $\Pr_{x \sim P_{\text{data}}}[x = 0] = \Pr_{x \sim P_{\text{data}}}[x = 100] = 0.5$. This data distribution ensures a close-form for the ground truth consistency function:

$$f^{\star}(x, t) := \begin{cases} 0 & \text{if } x < 50 \exp(-t) \\ 100 & \text{o.w.} \end{cases}.$$

We construct a perturbed $\hat{f}(\cdot, \cdot)$ accordingly:

$$\hat{f}(x, t) := \begin{cases} 0 & \text{if } x < a_t \\ 100 & \text{o.w.} \end{cases},$$

where the sequence $a_t$ satisfies: $\Pr_{x \sim P_t}[x < a_t] = 0.5 + 0.0001 t^2, \ \forall t$. This choice of $\hat{f}(\cdot, \cdot)$ makes sure:

$$\mathbb{E}_{\mathbf{x} \sim P_t}\left[\left\|\hat{f}(\mathbf{x}, t) - f^{\star}(\mathbf{x}, t)\right\|_2^2\right] = t^2.$$

This means $\hat{f}(\cdot, \cdot)$ satisfies the first statement of Lemma 2 with $\frac{\epsilon_{\text{cm}}^2}{\Delta \tau^2} = 1$.

We simulate three instantiations of $\{t_i\}_{i=1}^N$ defined in (5), i.e. the sequence of time steps for our multi-step sampling defined in (5):

- **our schedule:** the two-step schedule suggested by **Case study 1**. We also calculate the upper bound in (27) for comparison;
- **baseline 1:** design the sequence of sampling time steps by evenly dividing an interval;
- **baseline 2:** start with some $T$ and reduce it by half every step until reaching a small value.

In Figure 2, we plot the $W_2$ error in multi-step sampling. We present the revolution of $W_2$ error in a sampling time schedule on a single curve. Specifically, we plot each curve by:

$$\left(t_i, W_2(\hat{P}_0^{(i)}, P_{\text{data}})\right) \quad i = 1, \ldots, N.$$

Because the sampling time step $t_i$ decreases in the multi-step sampling by definition. We reverse the $x$-axis of the plot for presentation purposes.

**Observations:** This simulation result demonstrates that:

- Our upper bound is a reasonable characterization of the performance for the designed sampling time schedule.
- The two-step sampling time schedule suggested by **Case study 1** achieves comparable performance to the best result in the baseline methods but with a much smaller number of function evaluations;
- Running too many sampling time steps may degrade the sampling quality. The error increases for both baseline methods in the last few sampling steps.

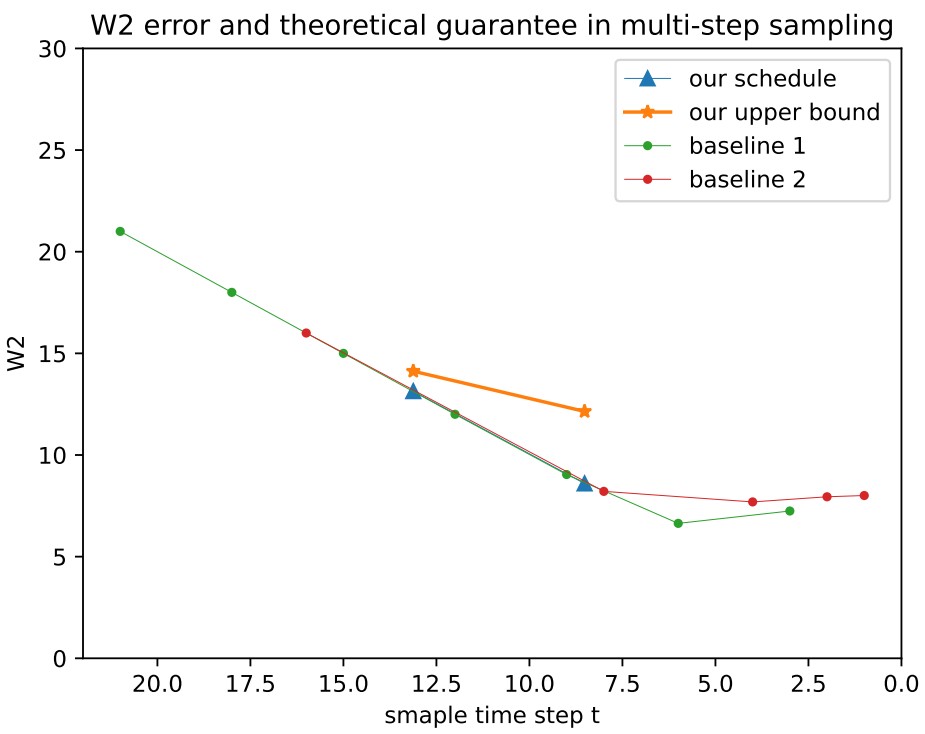

Figure 2: $W_2$ error in multi-step sampling.

