# OpenReview forum: "Convergence Of Consistency Model With Multistep Sampling Under General Data Assumptions"
_ICLR.cc/2025/Conference — Submitted to ICLR 2025_

### Official Review · Reviewer_rxyV · 2024-11-03

**Soundness:** 3
**Presentation:** 1
**Contribution:** 2
**Rating:** 5
**Confidence:** 3

**Summary:**

The paper studies the convergence of consistency models theoretically. They provide bounds on the Wasserstein distance between target and estimated data distribution using such models for as most general assumptions as they can (specifically, for bounded density support and light tail density). They transfer this to total variation distance under smoothness assumption, giving convergence guarantee. Moreover, they also derive theoretically the benefit of an additional sampling step in the specific case of Ornstein-Uhlenbeck process. They provide advance in the theoretical understanding of empirical observations about consistency models which are (i) an additional step can significantly improve the quality of the sample and (ii) more steps only bring limited gains.

**Strengths:**

The will of bringing theoretical claim about convergence of a method + analytical explanation of empirical observations is highly welcomed. It allows better understanding of flaws and successes of a method. Especially in the field of diffusion models, and specifically here consistency models which are SOTA methods for generative modeling. Consistency models aim to improve over classical diffusion, tackling the sampling efficiency. Bringing theory and explainability in convergence of such method is valuable.

- The paper base their claims on previous observations in the literature and try to improve in several ways.

- It is welcomed to see work that tries to provide theoretical claim with light assumptions, making them as most general as possible.

- Interpretability (in the case of multi-step sampling) is thrilling.

**Weaknesses:**

As a general comment, I found the goal laudable. However, the paper is really hard to follow and lacks clarity in its presentation and writing. I will try to pack similar remarks in thematic groups here below.

**General confusion**

The introduced notations are overly complex and sometimes confused.
- Line 50-71 : The inline paragraphs bold titles are weird. Why not using clear separated paragraphs? (I know that you may lack space, I propose getting rid of useless repetition below).

- Score-based generative model notations are hard to follow.

    - Is the switch between $P_t$ and $p_t$ needed?

    - Operator $\mathcal{D}$ is not useful and can be simplified by just using the noise schedule kernel and the distribution defined line 143.

    - Why $x_{t_i}$ and not just $x_i$? e.g. : line 165 where we have $x_{t_0} = x$, usually denoted $x_0$ in the literature. But on the other hand, you define a $\Delta \tau$ accounting for step between consecutive time, ... all of that can be simplified. I understood the paper but found it hard due to intertwined overly complex notations (at least understandable but making it hard to follow without making tons of back and forth in the text to check each piece). The time subscript is sometimes omitted, confusing also the reader concerning what we are talking about.

- The paper lacks structure and the reader is too often referred to other part of the paper. The authors introduce a lot of preliminaries 'useful for later', 'used in theorem x'... leading to a lot of repetition in the main text (in the abstract, the contributions, the theorem in themselves,...). It would be better to have more structured blocks. E.g. : Contributions can be shortened, more summarized, leaving the reader to the proofs in the main text.

- The consistency function introduced line 167-168 is used on distribution line 211. Of course the author precise **after** that the operation is now on distributions, but this further confuse the notations and highlight again a general confusion.

- It would be nice to introduce Wasserstein distance and total variation somewhere. Even in Appendix, it's ok. Indeed, In theorem 2 you use $W_2$ and then make a sort of connection with the KL line 187 that pops from nowhere. It lacks proper explanation.

- Lines 246-255 repeats a lot with preceding text.

**Results**

Analytical results are interesting. However they stick to theory and it's hard to identify their contribution in practice. Could you add, even for small toy problem, experiments that illustrate your findings. Indeed, you derive bounds given several assumptions. Use those for a toy example. It is crucial to link theory to practice to motivate the worth of your work. Experiments might also help identify potential yet to study problems/questions. Your results are sometimes specific (e.g. you stick to VP and VE SDE in case study 2) which is normal for theoretical derivations, please extend the scope of your work a bit. Even if you show that your conclusions do not hold at all in other setup, it might be valuable as later study can try to understand why? Or in the opposite, if your claims extend in other setup, it might raise the question of generalization of your work.

**Found typos**

I list below typos I detected, just to correct them (I might have missed some).

- Uppercase in contributions (lines 76-92).

- Space after a point line 169 **. At a high level ...**.

- Line 262: ~insteading~ -> instead of.

- Where is Theorem 1? At the beginning of page 6 you directly have Theorem 2 but there are no Theorem 1. Keep separated counting between Assumptions, Lemmas and Theorems.

- Line 367: ~forawrd~ -> forward.

- Line 368: ~nosie~ -> noise.

- Line 377: Without loss of generality preferred to not clear *W.l.o.g.*.

- Line 450: ~samling~ -> sampling.

**Suggestions**

- Remove the references to theorems in your contributions. Focus only to the key message of each of them and the rest will follow in the main text.

- line 165: remove the 'as a function of t'.

- Remove parenthesis around 'ground truth' line 166.

- Putting ':' inside a sentence often make it long and hard to follow (can be more structured). E.g. : lines 170-172

- Combine Theorem 2 and 3. Can't we see theorem 2 as a specific case of 3 when $C \to 0$?

For me, the paper is not ready yet. The main causes being the presentation and lack of illustrations.

**Questions:**

See Weaknesses.

---

> ### Author Response · Authors · 2024-11-25
>
> 1. Thank you very much for your suggestions. We have submitted a revision to resolve some of your comments.
> 2. $x_{t_i}$ vs $x_i$. Consistency model has both (a) a sequence of time steps for training, which goes from $0$ to $T$; (b) a sequence of time steps for inference, which decreases in the multi-step sampling process. For clarity, we directly use time $t$ as the subscript of $x$;
> 3. Could you please clarify the concerns on the scope of our work? We start with a **general** family of forward processes defined in Section 2 and present a sequence of theorems under this **general** setting. For illustration purposes, we instantiate our main result (the first theorem) with some VP and VE SDEs in the case studies and derive more interpretable upper bounds based on our main theorem. In fact, to the best of our knowledge, our results are much more general than previous works (Lyu et al. 2023, Li et al. 2024b, Dou et al. 2024) because they all focus only on VP SDEs but our results capture both VP and VE SDEs.
> 4. Please refer to Section G in the new version for empirical results.

---

> ### Comment · Reviewer_rxyV · 2024-11-27
> **Response**
>
> I have read the author's answer, and I will increase my score as part of my concerns have been addressed.
>
> However, I still find the practical contribution of the paper + the presentation (overall structure, ..) as detailed previously not clear. I understood your contributions (that you explicit again in your previous comment) but, for me, the impact of your paper seems limited as it is described now in the paper.
>
> For 'actionnable' input, I already gave explanations in my review. I think that a lot of notations + the structure is confusing and sometimes shadows the key messages. I think the paper would benefit from a refactoring, focused on concise, sufficient and clear notations. Some reviewers seem to share my thoughts about the presentation.
>
> For the new empirical study, I totally share eesS reviewer's opinion. I t should be included in the main text and better explained and introduced.

---

> > ### Author Response · Authors · 2024-12-01
> >
> > Thank you for raising your score! Could you please elaborate on your concerns regarding the impact of our work? We will do our best to address them.
> >
> > We once again appreciate your suggestion to improve the presentation of our paper. Since consistency model is heavy in notations, we are seeking a balance between rigor and clarity on the notations. We would definitely incorporate your suggestions in future revisions.
> >
> > In the meantime, we kindly refer you to our general response regarding the empirical study.

---

### Official Review · Reviewer_eesS · 2024-11-03

**Soundness:** 3
**Presentation:** 2
**Contribution:** 3
**Rating:** 6
**Confidence:** 3

**Summary:**

This paper studies the convergence of consistency models when the training process is approximately self consistent under the training distribution. This analysis involves mild data assumptions and applies to multi-step sampling for a family of forward processes. Sample quality guarantees are provided in terms of Wasserstein distance and total variation distance for target distributions that have particular properties. Finally, two multi-step sampling case studies are presented to demonstrate the implications of the results in this paper for two common forward processes.

**Strengths:**

- The paper provides a good overview of background and related work on score-based diffusion models and consistency models.
- The contributions in this paper regarding sample quality guarantees in Wasserstein distance and total variation distance appear to be novel and technically sound.
- The high-level ideas for the proofs of the main results in this paper are presented reasonably well in Section 4.

**Weaknesses:**

- While some of the implications and benefits of the results in this paper are presented as case studies from a theoretical standpoint in Section 3.3, there are no empirical/experimental results to accompany this discussion. The lack of such empirical results detracts somewhat from this paper, and adding these results would help to further validate the benefits of the authors’ results.
- To connect the results in this paper to real-world scenarios, it would be help for the authors to connect the data distribution properties used in Theorems 2 (bounded support) and 3 (light tail) and the data distribution smoothness assumption in Theorem 4 with some examples of real-world datasets commonly used to train consistency models that satisfy these properties/assumptions.

**Questions:**

It would be helpful for the authors to respond to the weaknesses pointed out above. If the authors don’t currently have experimental results that they can share, I suggest that they add such results to a future draft of this paper.

---

> ### Author Response · Authors · 2024-11-25
>
> 1. Our bounded support assumption is satisfied by the distribution of natural images because all images have finite pixels with ranges in $[0,1]$. The light tail and smoothness assumptions are mainly for theoretical purposes, both are common in theoretical works of diffusion models (Chen et al. 2023a, Chen et al. 2023b, Lee et al. 2023). Importantly, our results scale peacefully in those problem parameters.
> 2. Please refer to Section G in the new version for empirical results.

---

> > ### Comment · Reviewer_eesS · 2024-11-26
> > **Rebuttal response**
> >
> > I have read the authors' response to my review, as well their responses to the other reviewers, including the revised version of the paper. I thank the authors for their response to my concerns, as well as the concerns raised by the other reviewers. More specifically, I appreciate that the authors have added a new section G to the Appendix, with empirical results. However, the empirical results should be included in the main paper, rather than the appendix, and the presentation of the content in this section should be improved. For example, the discussion around Figure 2 is somewhat hard to follow. I also agree with some of the comments from reviewer rxyV regarding lack of clarify in the presentation of the paper. My rating remains a 6 (marginally above the acceptance threshold).

---

> > > ### Author Response · Authors · 2024-12-01
> > >
> > > Thank you very much for your feedback. We have made a minor revision to improve the structure of Section G. We would be happy to clarify any confusion. Additionally, we kindly direct you to our general response regarding the empirical results.

---

### Official Review · Reviewer_LeNY · 2024-11-06

**Soundness:** 2
**Presentation:** 3
**Contribution:** 3
**Rating:** 6
**Confidence:** 3

**Summary:**

This paper analyzes the distribution generated from the consistency model. It studies the upper bound of the total variation distance and Wasserstein distance between the generated distribution to true data distribution when the consistency model is approximately self-consistent. It then provides two examples that illustrate the main results.

**Strengths:**

The paper provides the upper bounds for the consistency model and discusses two examples to give the exact bounds. It studies the analytical aspect of the diffusion models and uses the results to provide insights into optimal choices for sampling. It contributes to the development of diffusion models.

**Weaknesses:**

The major concern is the tightness of the upper bounds discussed in the paper. It seems to me that given $R$ is never zero, the right-hand side of Equation (7) will be bounded away from zero. This might be because either $\hat{P}^{(N)}_0$ will not be asymptotically close to the true date distributions or the bound can be further tightened. Or is there a non-trivial condition that the right-hand side of Equation (7) would be small?

**Questions:**

a. It would be nice to also discuss the bounds when self-consistency holds.

b. The authors have provided two examples of case studies with specific upper bounds. It would be great to run the simulation on these two examples and see if the upper bound of calculated distance $W_2(\hat P_{t_{N}}, P_{\text{data}})$ matches the theories.

c. Is there a general rule of choosing optimal $N$?

---

> ### Author Response · Authors · 2024-11-25
>
> 1. The RHS of eq (7) would be small for a non-trivial set of problems. RHS is controlled by: the consistency error $\epsilon_{cm}$ and $\frac{\alpha_{t_1}^2}{\sigma_{t_1}^2}$. $\epsilon_{cm}$ is small for reasonable consistency functions. For now, let's assume $\epsilon_{cm}=0$ and focus on the other term. $\frac{\alpha_{t_1}^2}{\sigma_{t_1}^2}$ characterizes the convergence of the forward process to the Gaussian distribution, which goes to $0$ quickly for reasonable forward SDE. Please refer to our first technical Lemma in Section F for this connection. Indeed, the success of diffusion models relies on fast convergence: we add sufficient noise so that the forward marginal distribution is close to some Gaussian; we then approximate the marginal distribution with Gaussian and run the reverse process to generate samples. In OU-process (Case study 1), $\frac{\alpha_{t_1}^2}{\sigma_{t_1}^2} \approx \exp(-2t_1)$ so $W_2 = O(R^{1.5}\exp(-t_1/2))$, which goes to $0$ exponentially fast as $t_1\to\infty$; in VE process (Case study 2), $\frac{\alpha_{t_1}^2}{\sigma_{t_1}^2} \approx t_1^{-2}$ so $W_2 = O(R^{1.5}T^{-0.5})$. This means the term $\frac{\alpha_{t_1}^2}{\sigma_{t_1}^2}$ is ignorable when we choose a proper forward SDE with a reasonably large $t_1$.
> 2. In Case study 1, we show that two-step sampling should improve the sampling quality, but additional sampling steps will have diminishing improvement based on our upper bound. This is consistent with the finding in (Luo et al. 2023). The optimal choice of $N$ may vary for different forward processes.
> 3. Please refer to Section G in the new version for simulation results.

---

> ### Author Response · Authors · 2024-12-01
>
> Thank you for providing insightful feedback. The reviewer-author discussion period is ending soon. Could you please let us know if we have addressed all of your concerns? If there are no remaining issues, we would greatly appreciate it if you could consider adjusting your score accordingly.

---

### Official Review · Reviewer_Ruor · 2024-11-07

**Soundness:** 2
**Presentation:** 3
**Contribution:** 2
**Rating:** 5
**Confidence:** 3

**Summary:**

This paper gives the convergence of the consistency model multistep sampling procedure. The authors establish guarantees on the distance between the sample distribution and data distribution in terms of both Wasserstein distance and total variation distribution. The established upper bound requires only mild assumptions on the data distribution.

**Strengths:**

1. The theoretical results seem solid
2. This manuscript is well-written and easy to follow

**Weaknesses:**

1. The primary contribution stems from the utilization of a modest assumption. Nevertheless, this mild assumption still appears to have limitations in practical applications.
2. There is no empirical evaluation to verify the theoretical findings.

**Questions:**

1. It would be beneficial if the authors could illustrate the main contribution through a table.
2. Assumption 1 also relies on the distribution $P_{\tau_i}$ . It still seems to have limitations in practical applications. What advantages does it offer over other theoretical assumptions?
3.  It would be better if the authors could offer empirical evaluations to validate these theoretical findings.
4. The primary challenge lies in how to translate the approximate self-consistency measured under the training distribution into the quality of the generated samples. What technical breakthrough has been achieved to address this challenge?

---

> ### Author Response · Authors · 2024-11-25
>
> 1. Our main theorem requires two assumptions:
> (a) **bounded support of data distribution:** This assumption is satisfied by natural images because every image has finite pixels with value bounded in $[0,1]$;
> (b) **the approximate self-consistency (Assumption 1):** This assumption aligns with the loss function in the training process. The dependency on distribution $P_{\tau_i}$, the marginal distribution of the forward process, is fine since we can directly sample from it by adding noise to the image dataset. It's also common for theoretical works on diffusion models to make similar assumptions on the score functions;
> 2. Technical breakthroughs:
> (a) **avoid Lipschitz condition on consistency function:** one difficulty in translating the self-consistency property to sampling quality is how to account for the error of starting with Gaussian noise rather than the truth marginal distribution. Previous works (Lyu et al. 2023, Li et al. 2024b, Dou et al. 2024) assume Lipschitz condition on the ground truth consistency function. Lipschitz continuity can be used to analyze this error directly, but the unknown Lipschitz coefficient makes the result hard to interpret. We avoid making such assumption by using data-processing inequality;
> (b) **use the chain rule of KL to analyze multi-step sampling:** we need KL divergence on the starting distribution to apply data-processing inequality to avoid the Lipschitz assumption on the ground truth consistency function. However, W2 distance is a more natural quality metric for the generated data. It is challenging to translate between KL and W2 in the iterative process of multi-step sampling. To solve this issue, we apply the chain rule of KL to study how the mismatch (in terms of KL) in starting distribution changes during multi-step sampling and only translate KL to W2 in the last step;
> (c) **general forward processes:** previous works are limited to variance preserving forward processes, while we analyze a general family of forward processes that captures both variance preserving and variance exploding forward processes.
> 3. Please refer to Section G in the new version regarding **empirical evaluation**.
> 4. Please let us know if there are additional concerns about our assumptions or other aspects.

---

> ### Author Response · Authors · 2024-12-01
>
> Thank you for providing insightful feedback. The reviewer-author discussion period is ending soon. Could you please let us know if we have addressed all of your concerns? If there are no remaining issues, we would greatly appreciate it if you could consider adjusting your score accordingly.

---

### Author Response · Authors · 2024-12-01
**General response regarding experiments**

We thank all reviewers for your efforts in reviewing our paper. Several reviewers noted the lack of experiments in our original submission,  and we would like to address these concerns as follows:
1. **Moving simulation to the main text**: We added simulation results with a toy example in Section G in a recent revision. We hope this helps justify our theoretical results. However, we believe moving this section to the main text is unnecessary because it is purely for illustration purposes. Our primary contribution lies in providing a theoretical understanding of consistency models, not empirical validation. Including this simulation in the main text might shift focus away from our theoretical contributions.
2. **Absence of empirical result in a theory paper**: experiments are not always essential in theory-focused papers. Despite the tremendous amount of practical applications of score-based generative models, it is uncommon for a theoretical paper in this field to include experiments (See [1-6] below). The practical efficacy of diffusion and consistency models is already well-established, so we chose to focus on advancing the theoretical framework.
3. **Theoretical contributions**: we highlight our theoretical contribution here: (a) we relax the data assumptions in previous work by removing the Lipschitz condition on the consistency function; (b) we study a more general set of forward processes, which captures the widely-used variance exploding SDE; (c) we develop novel techniques to analyze multi-step sampling. All of these contribute to a more interpretable theoretical result. We respectfully ask the reviewers to evaluate our paper from this perspective.




[1] Chen et al. Improved analysis of score-based generative modeling: User-friendly bounds under minimal smoothness assumptions.

[2] Chen et al. Sampling is as easy as learning the score: theory for diffusion models with minimal data assumptions

[3] Dou et al. Theory of consistency diffusion models: Distribution estimation meets fast sampling

[4] Lee et al. Convergence of score-based generative modeling for general data distributions

[5] Li et al. Towards a mathematical theory for consistency training in diffusion models

[6] Lyu et al. Convergence guarantee for consistency models

---

### Meta-Review · Area_Chair_hMwy · 2024-12-23

**Metareview:**

A consistency model is proposed to converge to the true distribution of a diffusion model, so that data can be generated more efficiently. Although the approximation error is shown to be bounded, reviewers raised concern about the tightness of the bound. There are also criticism about lack of numerical experiments to support the theoretical claims.

**Additional Comments On Reviewer Discussion:**

A rough simulation is added to the appendix as per reviewers' request. Discussion about the theoretical contribution of the work is not able to convince the reviewers for higher ratings.

---

### Decision · Program_Chairs · 2025-01-22

Reject